# B cells suppress medullary granulopoiesis by an extracellular glycosylation-dependent mechanism

Eric E Irons[1], Melissa M Lee-Sundlov[2], Yuqi Zhu[3], Sriram Neelamegham[3], Karin M Hoffmeister[2], Joseph TY Lau[1]*

[1]Department of Molecular and Cellular Biology, Roswell Park Comprehensive Cancer Center, Buffalo, United States; [2]Blood Research Institute Versiti, Milwaukee, United States; [3]Department of Chemical and Biological Engineering, University at Buffalo, Buffalo, United States

**Abstract** The immune response relies on the integration of cell-intrinsic processes with cell-extrinsic cues. During infection, B cells vacate the marrow during emergency granulopoiesis but return upon restoration of homeostasis. Here we report a novel glycosylation-mediated crosstalk between marrow B cells and hematopoietic progenitors. Human B cells secrete active ST6GAL1 sialyltransferase that remodels progenitor cell surface glycans to suppress granulopoiesis. In mouse models, ST6GAL1 from B cells alters the sialylation profile of bone marrow populations, and mature IgD+ B cells were enriched in sialylated bone marrow niches. In clinical multiple myeloma, ST6GAL1 abundance in the multiple myeloma cells negatively correlated with neutrophil abundance. These observations highlight not only the ability of medullary B cells to influence blood cell production, but also the disruption to normal granulopoiesis by excessive ST6GAL1 in malignancy.
DOI: https://doi.org/10.7554/eLife.47328.001

*For correspondence:
joseph.lau@roswellpark.org

**Competing interests:** The authors declare that no competing interests exist.

## Introduction

Hematopoiesis generates the blood cells necessary for gas exchange, hemostasis, and immune defense. Cell-intrinsic developmental programs orchestrate these lineage decisions, but they are guided by systemic signals to convey the dynamically changing needs for specific cell types (*Lee et al., 2017*). Dysregulated communication of such extrinsic cues results in imbalanced blood cell production and can trigger pathologic processes including anemia, thrombocytopenia, inflammation, and autoimmunity (*Chovatiya and Medzhitov, 2014*; *Calvi and Link, 2015*). In malignancy, the disruption of normal differentiation within hematopoietic stem and progenitor cells by tumors can lead to insufficiencies in one or more blood cell lineages, a common complication (*Gabrilovich, 2017*).

The sialyltransferase ST6GAL1 is a glycan-modifying enzyme mediating the attachment of α2,6-sialic acids. Canonically, it resides within the intracellular ER-Golgi secretory apparatus, but there is also an extracellular blood-borne form (*Kim et al., 1971*; *Bernacki and Kim, 1977*; *Ip, 1980*). In addition to ST6GAL1, a number of other terminal glycosyltransferases are also present in systemic circulation (*Lee-Sundlov et al., 2017*). Fluctuating levels of blood-borne ST6GAL1 are associated with a wide array of conditions, especially metastatic cancers, where high levels of blood enzyme have been associated with poor patient outcomes (*Ip and Dao, 1978*; *Evans et al., 1980*; *Weiser et al., 1981*; *Berge et al., 1982*; *Dao et al., 1986*; *Cohen et al., 1989a*; *Magalhães et al., 2017*; *Rodrigues et al., 2018*). Early reports also associated elevated blood ST6GAL1 with systemic inflammation (*Kaplan et al., 1983*; *Jamieson et al., 1993*), atherosclerosis (*Gracheva et al., 1999*), Alzheimer's disease (*Maguire et al., 1994*), and alcohol-induced liver disease (*Malagolini et al.,*

*1989*; *Garige et al., 2006*; *Gong et al., 2007*; *Gong et al., 2008*). The physiologic contributions of extracellular ST6GAL1 in these diseases, however, remain poorly understood. We have hypothesized that secreted ST6GAL1 can access distant sites to modify circulating plasma components and surfaces of target cells that do not express ST6GAL1 (*Nasirikenari et al., 2014*; *Manhardt et al., 2017*). Previously, we observed that blood-borne ST6GAL1 can profoundly modify leukocyte differentiation by attenuating G-CSF dependent granulocyte production (*Dougher et al., 2017*) while promoting BAFF-dependent survival in B cells (*Irons and Lau, 2018*). In mouse models, circulatory ST6GAL1 insufficiency results in an exuberant granulocytic inflammatory response (*Nasirikenari et al., 2010*; *Nasirikenari et al., 2006*) that can be therapeutically ameliorated by intravenous infusion of recombinant ST6GAL1 (*Nasirikenari et al., 2019*). Extracellular, liver-derived ST6GAL1 is also a major determinant of serum immunoglobulin G sialylation, which activates anti-inflammatory pathways within innate immune cells through Fc receptors (*Jones et al., 2016*; *Jones et al., 2012*; *Pagan et al., 2018*). In the periphery, activated platelets can supply the necessary sugar donor-substrate to support such extrinsic glycosylation reactions (*Lee-Sundlov et al., 2017*; *Manhardt et al., 2017*; *Wandall et al., 2012*; *Lee et al., 2014*).

The principal source of extracellular ST6GAL1 in circulation is believed to be the liver (*Jamieson et al., 1993*; *Appenheimer et al., 2003*; *Lammers and Jamieson, 1986*), where ST6GAL1 expression is activated by glucocorticoids and IL-6 (*Jamieson et al., 1993*; *Jamieson et al., 1987*; *Wang et al., 1990*; *Dalziel et al., 1999*). However, B cells also robustly express ST6GAL1, which synthesizes the ligands for sialic acid-binding receptors CD22 and Siglec-G (*Wuensch et al., 2000*; *Müller and Nitschke, 2014*). Here, we report that hematopoietic lineage cells, particularly B cells, contribute to the extracellular pool of functional ST6GAL1 and the sialylation of non-self cells both in vitro and in vivo. B cells secrete functionally active ST6GAL1 that sialylates hematopoietic progenitors in co-culture to suppress granulopoietic differentiation. In mouse models, we observed a positive correlation between IgD+ B cells and richly α2,6-sialylated niches of the bone marrow. In bone marrow specimens of treatment-naïve human multiple myeloma, there was a striking negative association between marrow plasma cell ST6GAL1 expression and the prevalence of bone marrow neutrophils. Our study is the first to demonstrate that the liver is not the sole source of ST6GAL1 responsible for extracellular sialylation, and underscores a novel potential relationship between B lymphocytes and hematopoietic progenitors influencing neutrophil production in the marrow.

## Results

### Human B lymphoblastoid cells secrete enzymatically active ST6GAL1

B cell expression of ST6GAL1 is critical for B cell development and function secondary to engagement of the lineage-specific lectin CD22 with α2,6-sialic acid (*Hennet et al., 1998*). Although ST6GAL1 is expressed in multiple tissue and cell types, it is thought that secreted, extracellular ST6GAL1 is exclusively derived from the liver, as a hepatocyte conditional knockout of *St6gal1* results in vastly reduced serum α2,6-sialyltransferase activity (*Appenheimer et al., 2003*). In addition to hepatocytes, mature B cells strongly express ST6GAL1 (*Wuensch et al., 2000*), and numerous other cell types also express ST6GAL1 to varying degrees (*Dalziel et al., 2001*). The ability of non-hepatic cells to secrete functional ST6GAL1 to drive extrinsic sialylation has not been formerly studied.

We have recently analyzed the expression of ST6GAL1 within bone marrow and splenic B cell populations in mice and found that maximal ST6GAL1 expression occurred in early transitional and mature stages of development (*Irons and Lau, 2018*). However, it is unclear if the ST6GAL1 expressed in B cells is also actively released into the environment. In order to assess if human B cells are capable of secreting ST6GAL1, we analyzed four B lymphoblastoid cell lines derived from multiple stages of differentiation. *ST6GAL1* mRNA expression was detectable in all cell lines except myeloma line RPMI 8226, with highest expression observed in the Burkitt lymphoma line Louckes (*Figure 1a*). Since Louckes is a germinal center B cell derivative, this observation is consistent with our previous observations that BCR activation induces ST6GAL1 expression (*Wang et al., 1993*). Expression of the β-site amyloid precursor protein-cleaving enzyme 1 (BACE1), thought to be required to liberate ST6GAL1 from its N-terminal membrane anchor prior to secretion

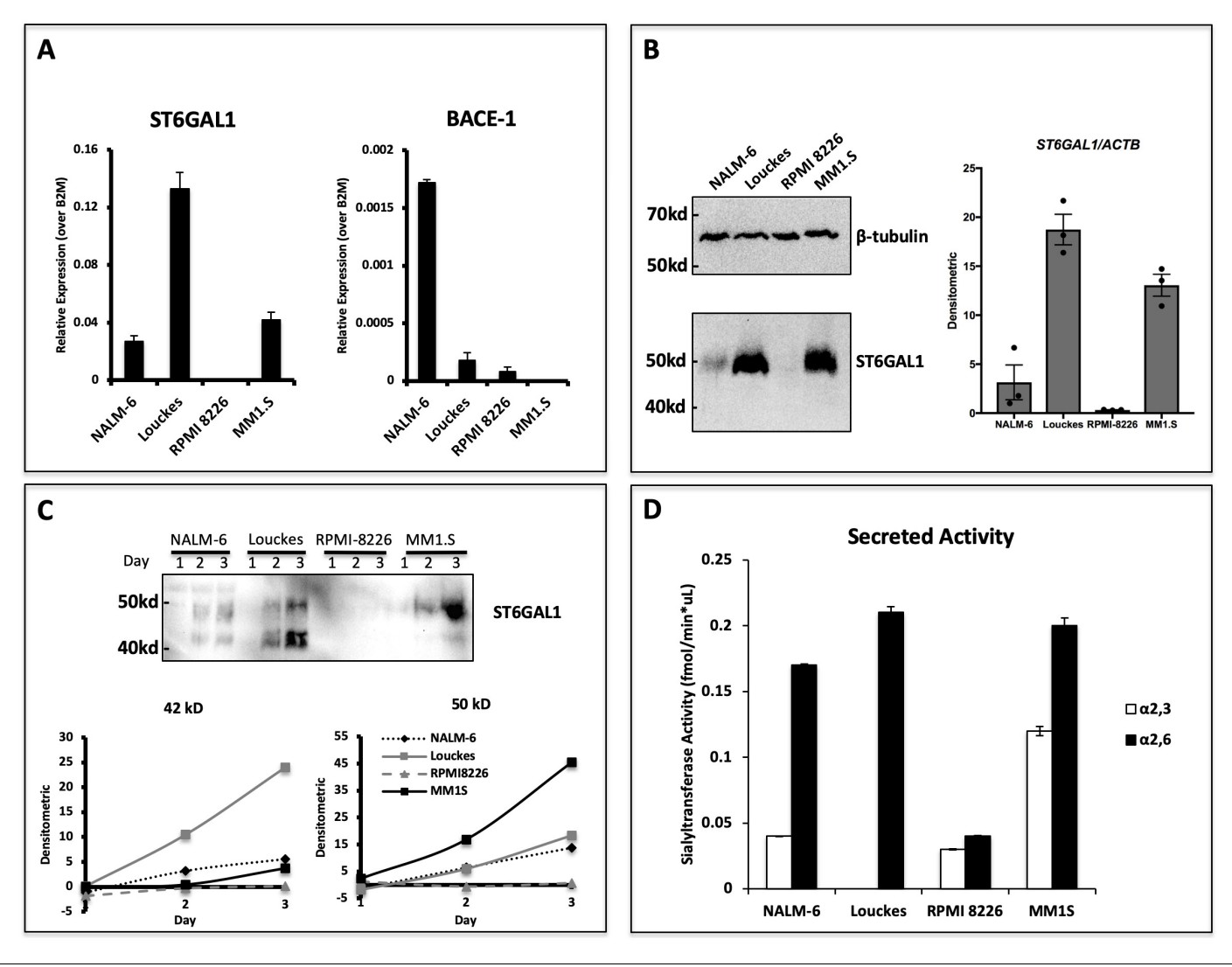

**Figure 1.** Human B lymphoblastoid cells secrete ST6GAL1. Human lymphoblastoid cell lines derived from the pre-B (NALM-6), germinal center (Loucks) and plasma cell (RPMI 8226 and MM1.S) stages were profiled for ST6GAL1 expression and secretion. (A) RT-qPCR analysis of *ST6GAL1* and beta-secretase BACE1 mRNA (n = 3 replicates) (B) Total ST6GAL1 protein analyzed by western blot (left) and quantified (right, n = 3). (C) Protein levels of ST6GAL1 in the serum-free conditioned medium of cell cultures 1–3 days after plating 106 cells/ml, analyzed by western blot (top) and quantified for 50 kD and 42 kD sizes (bottom). (D) Sialyltransferase activity in conditioned medium, relative to media only control, was determined by incorporation of [3H]NeuAc onto Gal(β4)GlcNAc-O-Bn acceptor substrate. [3H]-Labeled products were separated by SNA-agarose chromatography into [3H]NeuAc-α2,6-Gal(β4)GlcNAc-O-Bn α2,6- (SNA binding) and [3H]NeuAc-α2,3-Gal(β4)GlcNAc-O-Bn (SNA non-binding) fractions. The data shown are representative of multiple experiments with similar results.

DOI: https://doi.org/10.7554/eLife.47328.002

(*Kitazume et al., 2001*; *Deng et al., 2017*), was detected within all lines except the myeloma cell MM1.S (*Figure 1a*). The cellular content of ST6GAL1 protein, assessed by western blot of total cell lysates, essentially followed *ST6GAL1* mRNA levels. A possible exception was MM1.S cells, which expressed more ST6GAL1 protein than expected from *ST6GAL1* transcript levels. All cells examined expressed, as expected, the cellular full-length ST6GAL1 form of 50 kDa. (*Figure 1b*). To assess the ability of the B lymphoblastoid cell lines to secrete functional ST6GAL1, the cells were seeded in serum-free medium for 3 days, and ST6GAL1 released into the medium was analyzed by western blot and assayed for sialyltransferase activity. All cell lines, except RPMI-8226, released measurable ST6GAL1 protein in a time-dependent manner into the medium (*Figure 1c*). B cells (NALM-6, Loucks) expressing BACE1 secreted the expected 42 kDa soluble form of ST6GAL1, consistent with

the proteolytic liberation of the soluble catalytic active domain from the full-length protein by BACE1. Cells also released a 50 kDa form, consistent in size with the full-length ST6GAL1. MM1.S cells, which do not express BACE1, released predominately the 50 kDa form. To the best of our knowledge, the release of unprocessed, full-length ST6GAL1 has never been reported. The putative identity of the large 50 kDa form and its potential biologic significance are not explored further here. Enzymatic assay confirmed that all released ST6GAL1 was catalytically active, regardless of the larger size observed particularly in MM1.S (*Figure 1d*). Together, these results demonstrate that human B cell lines can release ST6GAL1 in vitro.

Theoretically, extracellular ST6GAL1 can enzymatically reconstruct sialic acid on cell surfaces in the presence of a sialic acid donor substrate (*Lee et al., 2014*). In order to determine if B cell secreted ST6GAL1 is capable of extrinsically remodeling cell surfaces, we applied concentrated conditioned medium from Louckes cells (concentrated ~20X) to sialidase-pretreated human hepatoma HepG2 cells (*Figure 2a*). The target HepG2 cells were rendered metabolically inert by formalin fixation to disable cell endogenous capacity to regenerate the cell surface sialyl glycans. Louckes conditioned media alone was insufficient to restore HepG2 cell surface SNA reactivity, but required the presence of the sugar donor substrate, CMP-Sia. The ability of Louckes conditioned media to restore HepG2 surface SNA reactivity in the presence of CMP-Sia could be reproduced in HepG2 cells in suspension, as analysed by quantitative flow cytometry (*Figure 2b and c*). These results indicate that the sialyltransferase secreted by B cells is enzymatically active and capable of extrinsic sialylation of cell surface glycans when supplemented with the sugar donor.

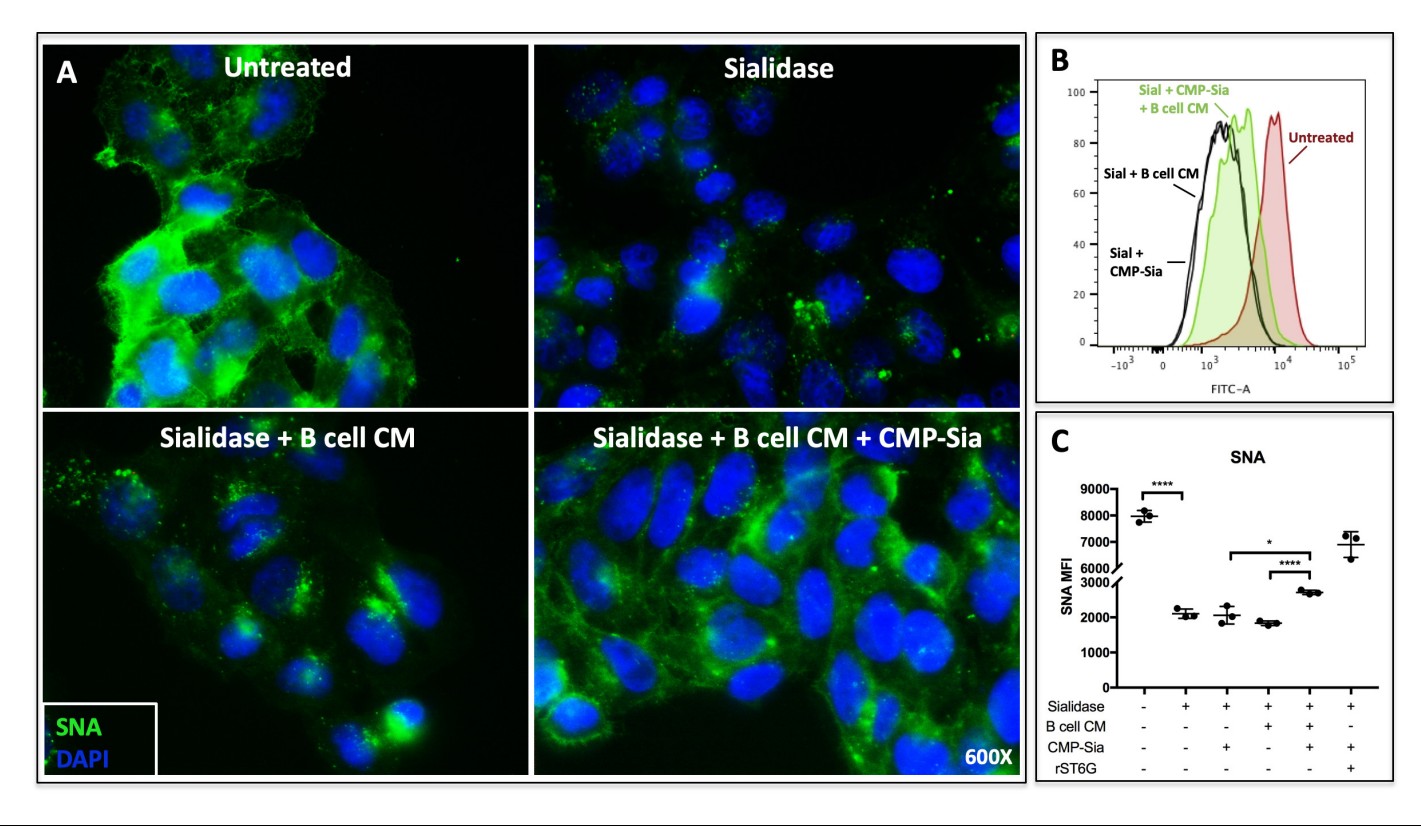

**Figure 2.** B cell conditioned medium extrinsically restores SNA reactivity of target cells. (**A**) HepG2 human liver cells were grown on glass cover slides and fixed (10 min in 5% formalin) to disable endogenous metabolism. Cells were treated with *C. perfringens* sialidase C (Roche) for 1 hr at 37C to remove cell surface sialic acid, then incubated with concentrated (~20X) B cell conditioned medium (CM) from Louckes grown in serum-free medium, in the presence or absence of 0.05 mM CMP-sialic acid. Representative images of cell surface sialylation, as indicated by SNA lectin stain, are shown. HepG2 cells in suspension were subjected to the same treatments and analyzed by flow cytometry for SNA reactivity. SNA reactivity of nucleated cells is shown (**B**) by representative histogram and (**C**) as average mean fluorescence intensity of biological replicates.
DOI: https://doi.org/10.7554/eLife.47328.003

# B cell ST6GAL1 sialylates hematopoietic progenitors to suppress granulopoiesis in co-culture

Mice deficient in circulatory ST6GAL1 have exaggerated neutrophilia inducible by various inflammatory stimuli (*Nasirikenari et al., 2010*; *Nasirikenari et al., 2006*; *Appenheimer et al., 2003*). Supplementation of recombinant ST6GAL1 is sufficient to blunt development of G-CSF and IL-5 dependent granulocytic colonies from whole bone marrow cells in vitro (*Jones et al., 2010*). In vivo, elevation of blood ST6GAL1 reduces bone marrow neutrophil abundance and neutrophilic inflammation secondary to sialylation of the multipotent granulocyte/monocyte progenitor (GMP). Ex vivo sialylation of GMPs potently inhibits G-CSF induced STAT3 signaling to reduce expression of myeloperoxidase, C/EBP-α, Gr-1, and ultimately, neutrophil production (*Dougher et al., 2017*). The data presented here suggest that B cells secrete enzymatically active ST6GAL1 capable of sialylating cell surfaces. Early B cells, mature B cells, and antibody-producing plasma cells represent a significant fraction of bone marrow cells. Therefore, we hypothesized that B cell-derived extrinsic ST6GAL1 participates in the modification of neighboring hematopoietic stem and progenitor cells (HSPC) to control granulopoiesis.

To test this, human B lymphoblastoid cell lines were co-cultured with c-kit+ *St6gal1*KO murine HSPCs and supplemented with SCF, IL-3, TPO, Flt-3, G-CSF, and CMP-sialic acid for 3 days. In order to resolve the two populations by flow cytometry, murine cells were labeled with the plasma membrane dye CellTrace Violet (*Figure 3a*). Co-cultures were seeded at 1:1 and 4:1 ratio of B cells to HSPCs. Co-culture of HSPC with B cells resulted in modifications to their cell surface sialic acid levels, as measured by reactivity towards the lectins *Sambucus nigra* (SNA) and *Maackia amurensis II* (MAL-II), which recognize α2,6- and α2,3-sialic acids, respectively (*Figure 3b*). The ST6GAL1 secreting B cell lines, NALM-6, Louckes, and MM1.S, increased cell-surface SNA reactivity of HSPC in a dose-dependent manner. RPMI-8226 cells, which do not release ST6GAL1, did not raise SNA reactivity on the co-cultured HSPCs. In contrast, cell surface MAL-II reactivity had no apparent correlation with ST6GAL1 status in the co-cultured B cells.

We assessed expression of CD11b and Gr-1, markers of granulocytic differentiation, on the murine HSPC (*Figure 3c*). Although CD11b is expressed before Gr-1 during granulocyte development in vivo, Gr-1 expression can precede CD11b during in vitro culture. We analyzed total expression of Gr-1 after co-culture and observed that ST6GAL1 expressing B cells were able to modify expression of Gr-1 on the murine HSPCs. Co-cultured murine HSPCs exhibited a significant negative correlation between Gr-1 expression and increasing cell surface SNA reactivity ($r^2 = 0.66$, p<0.0001) (*Figure 3d*). A similar correlation was not observed between pan-myeloid marker CD11b and SNA reactivity (*Figure 3d*), suggesting a specific effect on granulocytes, in contrast to total myeloid cells. We also analyzed the frequency of CD11b-/Gr-1- murine HSPCs, indicative of less-differentiated cells that had yet to commit to the myeloid or granulocyte lineage (*Figure 3e*). Consistently, there was a significant, dose-dependent increase in this undifferentiated population in co-cultures with *ST6GAL1* expressing B cell lines, suggesting that the presence of ST6GAL1 is correlated with maintenance of a less differentiated state. Since neither the Ly6C or Ly6G epitope of Gr-1 is predicted to contain any N-linked glycans, this relationship is unlikely a direct result of altered antibody binding due to epitope sialylation. Collectively, our data are consistent with a role of B cell-derived ST6GAL1 in the sialylation and differentiation of co-cultured *St6gal1*KO HSPC. However, patterns in HSPC SNA reactivity do not correlate perfectly with B cell ST6GAL1 expression in our data.

To establish definitively that the altered HSPC differentiation was due specifically to ST6GAL1 expression, two independent *ST6GAL1*KO genetically-modified MM1.S cell lines were generated using a CRISPR/Cas9 approach. These MM1.S modified lines, Cr3 and Cr4, express and secrete vastly reduced amounts of ST6GAL1 (see *Figure 3—figure supplement 1*). In co-cultures with murine *St6gal1*KO HSPC, *ST6GAL1* deletion in MM1.S cells abrogated the dose-dependent increase in SNA reactivity on murine HSPCs (*Figure 3f*, left). Furthermore, loss of *ST6GAL1* also attenuated the dose-dependent suppression of Gr-1 expression by co-cultured MM1.S cells (*Figure 3f*, right). Together, these observations implicate B cell secreted ST6GAL1 in the α2,6-sialylation and suppression of granulocyte differentiation in co-cultured HSPCs.

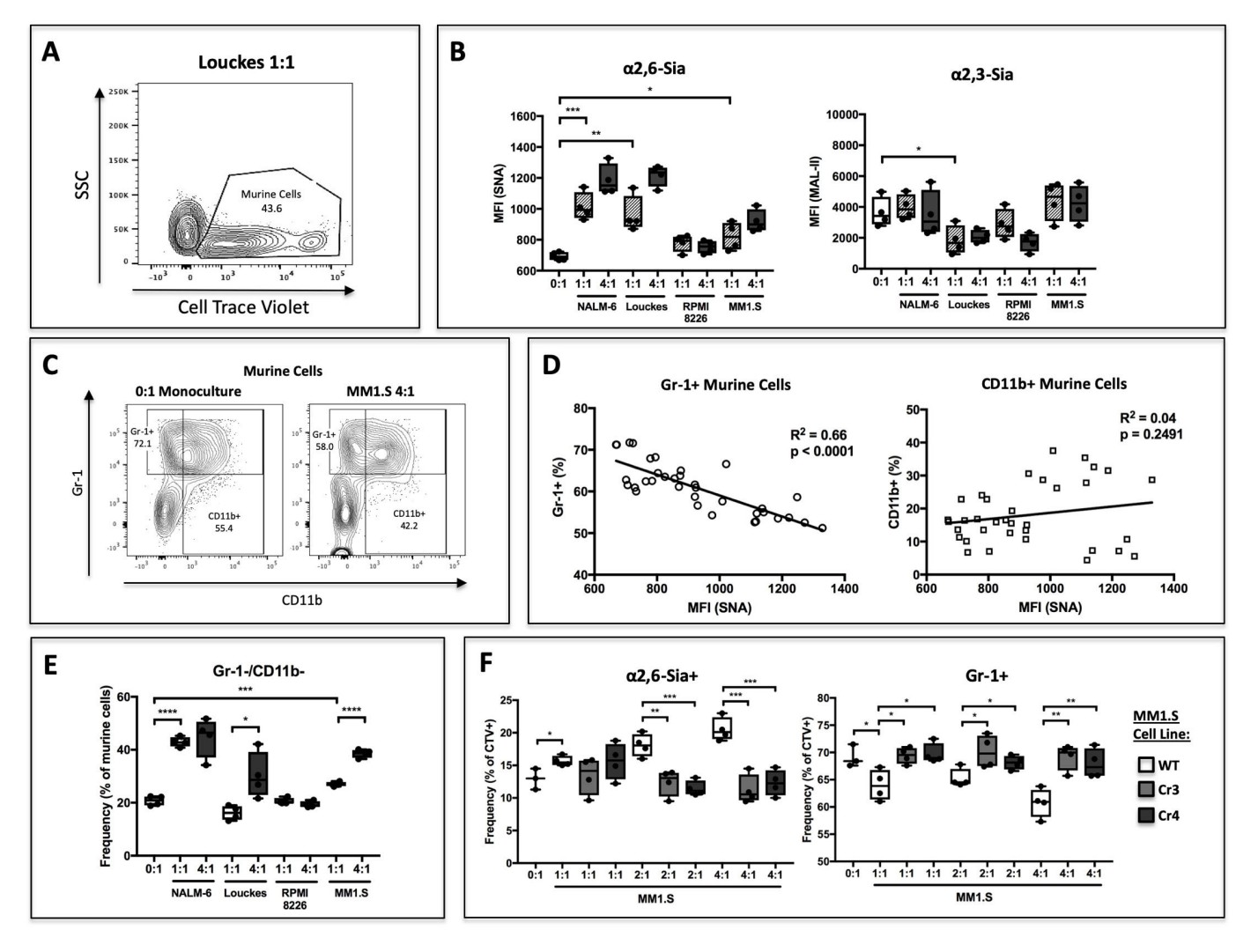

**Figure 3.** B cells modify HSPC SNA reactivity and Gr-1 expression in co-culture. Human B lymphoblastoid cell lines were co-cultured with CellTrace Violet-labeled murine *St6gal1*KO c-kit+ bone marrow cells for 3 days with SCF, IL-3, G-CSF, TPO, and Flt-3 at indicated ratios of 1:1, 2:1, or 4:1. (**A**) Resolution of B cells and murine HSPCs by flow cytometry, with HSPCs staining positive for CellTrace Violet. (**B**) Levels of SNA and MAL-II reactivity on the HSPCs in monoculture or co-culture with indicated B cells. (**C**) Flow cytometric separation of CD11b+ and Gr1+ cells from murine HSPCs, after 3 days of monoculture (left) or co-culture (right) with MM1.S myeloma cells at 4:1 ratio. (**D**) Correlation between murine cell SNA reactivity and frequency of Gr-1+ or CD11b+ cells (expressed as % of total CellTrace Violet+ cells). (**E**) Frequency of CD11b-/Gr-1- undifferentiated murine cells after co-culture. (**F**) Frequency of SNA+ and Gr-1+ murine cells after co-culture with genetically modified MM1.S cell lines (Cr3 and Cr4) with targeted *ST6GAL1* knockout by CRISPR/Cas9. Data are from a single experiment representative of three individual experiments, with n = 4 technical replicates per condition *p<0.05, **p<0.01, ***p<0.001 by student's T-test.

DOI: https://doi.org/10.7554/eLife.47328.004

The following figure supplement is available for figure 3:

**Figure supplement 1.** Genetic Modification of ST6GAL1 in Multiple Myeloma Cell Lines.
DOI: https://doi.org/10.7554/eLife.47328.005

## B cells secrete ST6GAL1 to modify non-self hematopoietic cells in vivo.

Our data demonstrate that human B lymphoblastoid cells can release functional ST6GAL1 to modify the glycosylation and granulopoietic differentiation of co-cultured hematopoietic progenitors in vitro. After neutrophils, B cells are the second most abundant lineage of hematopoietic cells in the marrow, and our early data indicated that they express significant ST6GAL1 (*Wuensch et al., 2000*). To assess if hematopoietic cells are a significant source of extracellular ST6GAL1 in vivo, wild type

(C57BL/6) or *St6gal1*KO whole bone marrow was transplanted into irradiated recipients. The recipients were *St6gal1*KO/μMT and thus deficient in ST6GAL1 and B cells due to loss of the heavy chain of IgM (*Ighm-/-*). Thus, all ST6GAL1 present in extracellular compartments (e.g. blood) must originate from donor hematopoietic cells (*Figure 4a*). Blood α2,3-sialyltransferase activity, which did not depend on ST6GAL1, was unchanged and remained within the limits of resting WT (native C57BL/6) animals (*Figure 4b*). In contrast, blood accumulation of α2,6-sialyltransferase activity against Gal(β4)

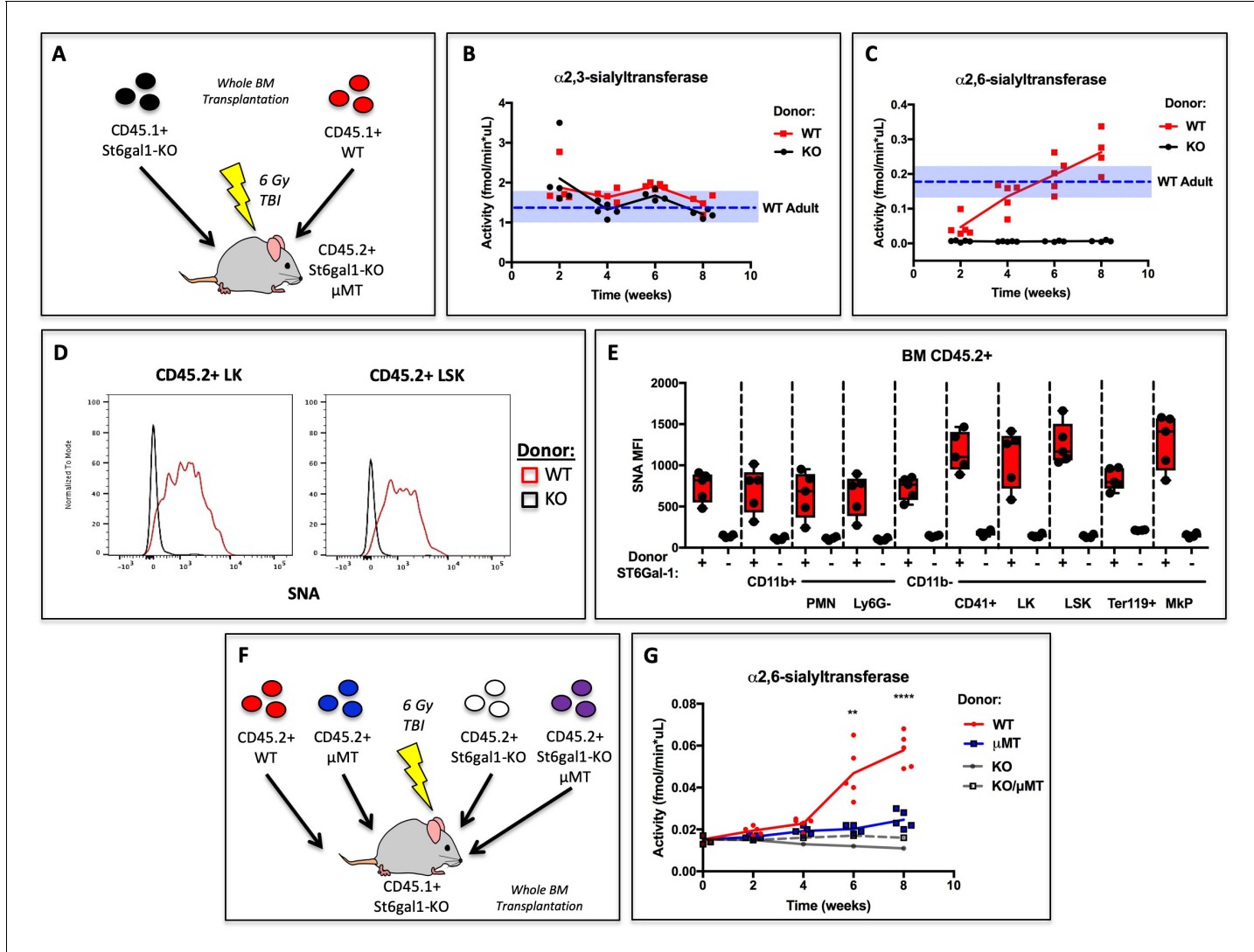

**Figure 4.** Hematopoietic Cells Supply Extracellular ST6GAL1 for Extrinsic Sialylation in vivo. (**A**) CD45.1 + *St6*gal1 sufficient (WT) or deficient (KO) whole bone marrow was used to reconstitute irradiated CD45.2+/*St6gal1*KO/μMT mice. (**B**) α2,3 and (**C**) α2,6-sialyltransferase activity was quantified in serum of bone marrow chimeras at indicated time points. (**D**) Representative histograms of SNA-reactivity are shown for Lin-/c-kit+/Sca-1- (LK) and Lin-/c-kit+/ Sca-1+ (LSK) progenitor compartments in the bone marrow, 8 weeks post-transplant. (**E**) Mean fluorescence intensity of SNA in CD45.2+ recipient bone marrow cell subsets was quantified by flow cytometry (n = 5). All cell types were significantly (p<0.01) different between WT and KO donor chimeras by student's T-test. (**F**) CD45.2+ WT, μMT, *St6gal1*KO, or *St6gal1*KO/μMT bone marrow was used to reconstitute irradiated CD45.1+ *St6gal1*KO recipients. (**G**) α2,6-sialyltransferase activity was quantified in serum of bone marrow chimeras at indicated time points. Statistical significance is indicated in comparisons of WT and μMT cohorts for student's T-test (**p<0.01, ****p<0.0001).

DOI: https://doi.org/10.7554/eLife.47328.006

The following figure supplement is available for figure 4:

**Figure supplement 1.** Chimerism of Analyzed CD45.2+ Host cells.
DOI: https://doi.org/10.7554/eLife.47328.007

GlcNAc-o-Bn acceptor was dependent on re-establishment of *ST6GAL1*-competent hematopoietic cells, and increased and even surpassed the baseline for resting WT mice by week 8 (*Figure 4c*).

In order to distinguish between donor and recipient hematopoietic cells, we utilized CD45.1 + mice as donors, and CD45.2+/*St6gal1*KO /μMT mice as recipients. Presence of CD45.2+ host derived cells was comparable between experimental groups (see *Figure 4—figure supplement 1*). Amongst the CD45.2+ residual host-derived CD11b+ myeloid, c-kit+ hematopoietic progenitors, and CD41+ megakaryocyte lineage cells, all of them unable to express their own ST6GAL1, increases in SNA reactivity were noted. Particularly, progenitor Lin-/c-kit+/Sca-1- (LK) and Lin-/c-kit+/Sca-1+ (LSK) populations, as well as CD41+ megakaryocyte lineage progenitors, were dramatically modified by cell non-autonomous extrinsic ST6GAL1, in comparison to CD11b+ myeloid cell subtypes and Ter119+ erythrocyte progenitors. Representative data depicting LK and LSK cell SNA reactivity are shown in *Figure 4d*. Quantitative data depicting all analyzed bone marrow cell types are shown in *Figure 4e*. These obeservations demonstrate that hematopoietic-derived ST6GAL1 alters the sialylation of non-self cells, and further suggest target-dependent bias of the extracellular ST6GAL1 towards CD11b-neg cells. These data demonstrate that hematopoietic cells alone, without contribution from the liver the canonical source of circulating ST6GAL1, were sufficient not only to maintain baseline blood ST6GAL1, but also to confer cell surface SNA-reactivity onto other cells that were unable to express their own ST6GAL1.

In order to assess definitively the contribution of B cells to extrinsic ST6GAL1 in vivo, the bone marrow transplantation approach was expanded to include comparisons between donors capable or incapable of reconstituting the B lineage (i.e WT or μMT). Comparisons were also extended to donors that were *ST6GAL1*-deficient and either B cell-intact or B cell-absent (i.e. *St6gal1*KO or *St6gal1*KO/μMT). All recipients were *St6gal1*KO and subjected to full body irradiation before transplantation (*Figure 4f*). The data show that ST6GAL1 accumulated in the blood of the chimeras reconstituted with WT marrow, but not in in chimeras reconstituted with μMT marrow. Strikingly, almost all (roughly 75%) of the increase in blood ST6GAL1 activity could be attributed to B cells. The parallel comparison, between *St6gal1*KO and *St6gal1*KO/μMT donors, did not show any detectable increase in serum ST6GAL1 activity (*Figure 4g*). Importantly, since both comparisons differ only in the presence or absence of B cells, and only differ from each other in the expression of *St6gal1*, the difference in blood ST6GAL1 activity can be attributed to ST6GAL1 expression in B cells. Thus, the data demonstrate that B cells directly express and release ST6GAL1 into the blood to restore extracellular levels in the absence of host ST6GAL1 expression.

We visualized SNA reactivity within the chimeric marrow of B cell deficient, *St6gal1*KO/μMT recipients reconstituted with WT marrow cells, in which both ST6GAL1 and mature, IgD+ B cells can only come from the donor (as in *Figure 4a*). The formalin-fixed and frozen whole femurs from these chimeras had strikingly patchy areas of SNA reactivity across the bone marrow (*Figure 5a*). In contrast, uniform SNAreactivity was observed in chimeras where the recipients were ST6GAL1-competent, for example μMT mice. This observation suggested that the spatially punctate distribution of marrow α2,6-sialylation results only when hematopoietic cells are the sole source of ST6GAL1. The distribution of α2,6-sialyl structures, as measured by SNA, was analyzed in the *St6gal1*KO/μMT recipients after engraftment with wild-type hematopoietic cells. Areas representing high, medium, and low reactivity towards SNA, designated as Regions of Interest (ROIs), were selected for analysis. ROIs were selected based on the relative homogeneity of SNA staining, rather than on region size or number of cells present. Four such ROIs are outlined in white in a single femur, with observed SNA staining intensity of Region 1 > Region 4 > Region 3 > Region 2 (*Figure 5b*, top). The whole femur was further stained for IgD to identify donor-derived mature recirculating B cells (*Figure 5b*, bottom). Quantitative analysis within four recipients demonstrated that the number of IgD+ B cells was modestly positively correlated with the SNA reactivity of ROIs (*Figure 5c*, $R^2$ = 0.2448, p<0.05). Distinctly, SNA+ cells were not limited to the IgD+ B cells, but often encompassed groups of cells in their vicinity, consistent with their extrinsic modification. Collectively, this suggests that some of the variation in bone marrow niche α2,6-sialylation can be attributed to the presence of mature IgD+ B cells.

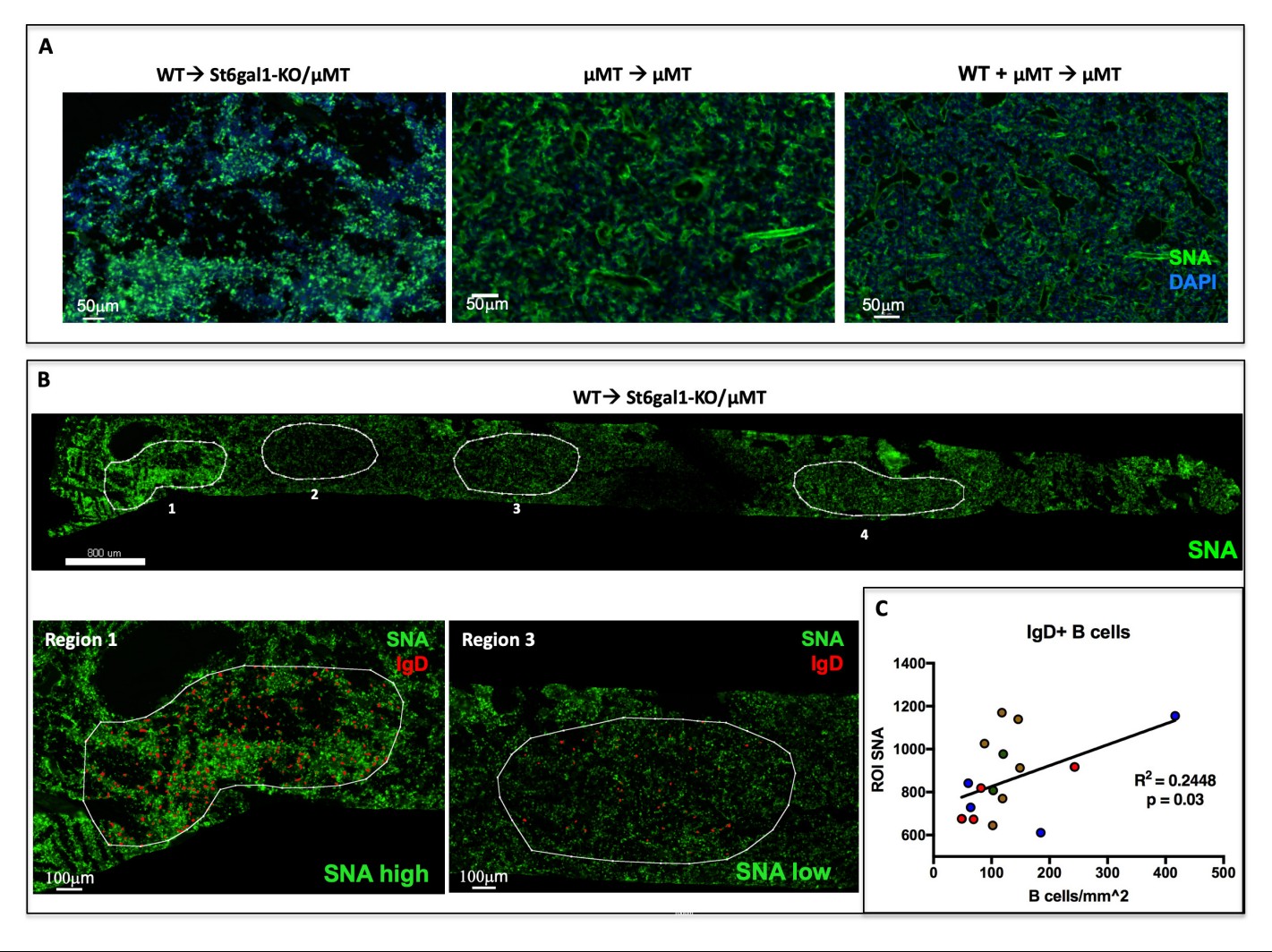

**Figure 5.** IgD+ B cells are Enriched in Regions of High α2,6-Sialylation within the Bone Marrow. (**A**) Wild-type or µMT bone marrow was allowed to reconstitute irradiated µMT or *St6gal1*KO/ µMT mice for 8 weeks. Upon sacrifice, whole femurs were fixed, frozen, and sectioned for immunofluorescence staining. (**B**) Heterogeneous SNA reactivity was observed and indicated regions of interest (ROI) were created with differing SNA reactivity (top). Mature B cells (IgD+) were identified within ROIs (bottom panels). (**C**) Correlation of overall SNA reactivity with abundance of IgD+ B cells in ROIs from chimeras. Data are derived from four biological replicates.

DOI: https://doi.org/10.7554/eLife.47328.008

## ST6GAL1 expression in human multiple myeloma negatively correlates with bone marrow neutrophil abundance

B cells exist within the bone marrow medullary spaces at several stages during development. Whereas early B cells occupy the niche until successful arrangement of a functional BCR, mature B cells freely recirculate between the marrow, blood, and lymphoid tissues. In contrast, plasma cells can remain for years within the bone marrow to maintain systemic titers of protective antibodies, and are thought to occupy a specialized niche in proximity to megakaryocytes, eosinophils, and soluble pro-survival factors (*Wilmore and Allman, 2017*). In order to understand if B cell-derived ST6GAL1 is able to perturb normal HSPC development into mature leukocytes in a clinically relevant setting, we performed histological analyses of human bone marrow samples from freshly diagnosed, treatment-naïve patients with a plasma cell dyscrasia, multiple myeloma. Samples from only a limited number of patients (n = 15) were available, and these were examined. Because of the clonal origin of multiple myeloma, the ST6GAL1 expression of the neoplastic plasma cell could be assessed.

Moreover, the high occupancy of the bone marrow by the neoplasm, which in some cases approached 60%, allowed for an assessment of the consequences of pathologically elevated ST6GAL1 within the bone marrow microenvironment.

Paraffin-embedded bone marrow sections were stained for ST6GAL1 using a DAB reagent. Plasma cell-specific expression of ST6GAL1 was assessed according to intensity (1-5) and frequency (0–100%) by counting five groups of ten cells each in at least five fields of view per patient. The product of intensity and frequency is referred to as 'ST6GAL1 score'. ST6GAL1 expression was highly heterogeneous between patients, and varied from nearly completely absent to intense expression in 100% of examined cells (*Figure 6a*). The expression of ST6GAL1 in tumor cells was not associated with altered patient survival (*Figure 6—figure supplement 1a*). However, whereas the plasma cell burden and abundance of segmented neutrophils varied completely independently (*Figure 6—figure supplement 1b*, $r^2 = 0.001$, p=0.88), we observed a striking relationship between plasma cell ST6GAL1 expression and segmented neutrophils. When ST6GAL1 score was compared to the frequency of granulocyte lineage cells, as assessed by a trained clinical pathologist, we identified a strong negative correlation between ST6GAL1 score and presence of segmented neutrophils

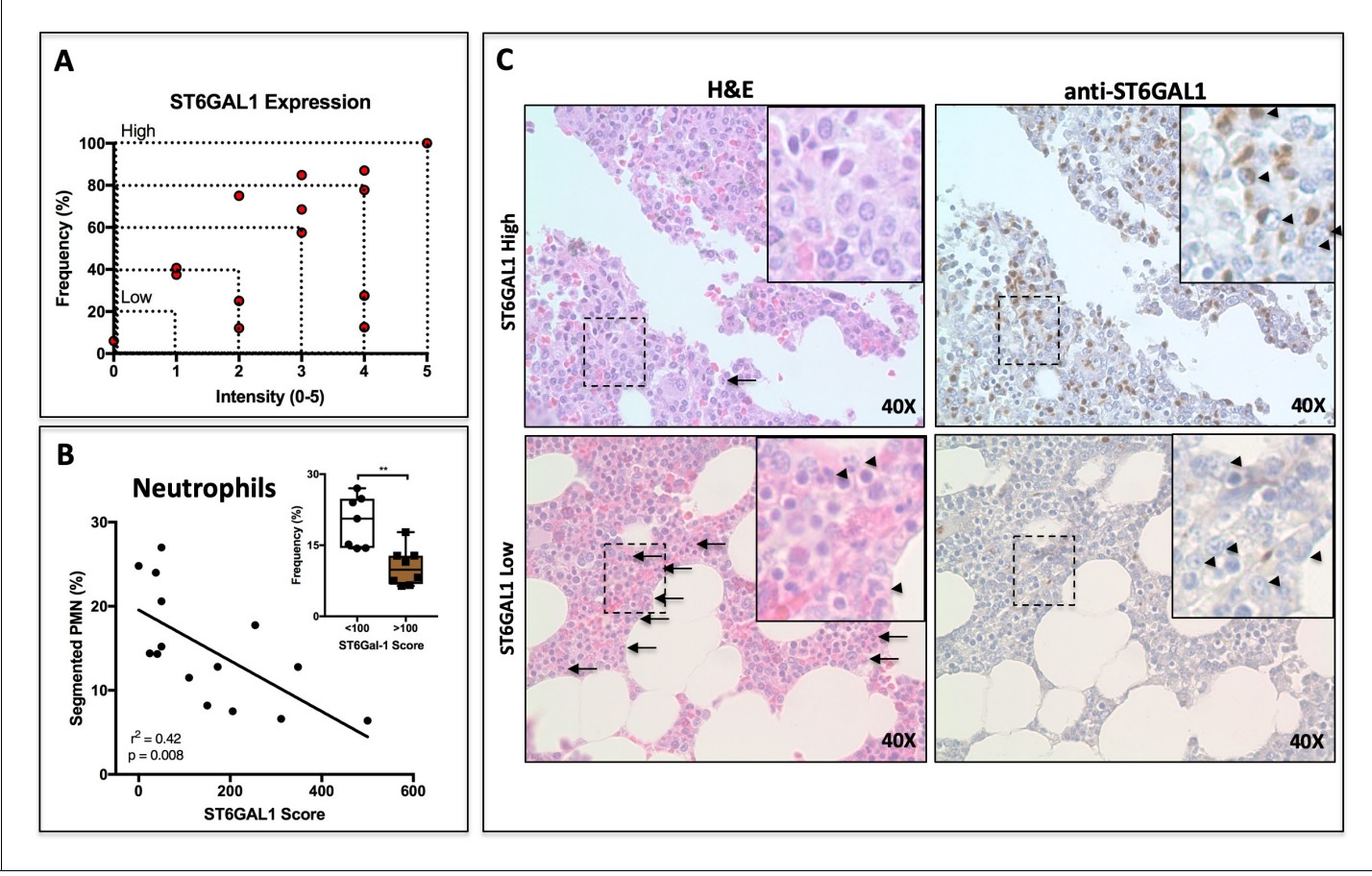

**Figure 6.** ST6GAL1 Expression in Human Multiple Myeloma Cells Correlates Negatively with Bone Marrow Neutrophil Abundance. (**A**) Quantification of ST6GAL1 expression in bone marrow histological specimens from treatment-naive multiple myeloma patients (n = 15). (**B**) Negative correlation between *ST6GAL1* expression and frequency of bands and segmented PMNs (p<0.01). Stratification of patients into low and high ST6GAL1 expression was predictive of abundance of segmented neutrophils (**\*\***p<0.01, student's T-test). (**C**) Representative data from one patient with high and one patient with low myeloma *ST6GAL1* expression, showing H and E with neutrophils indicated (left, arrows or arrowheads), and ST6GAL1 stain with myeloma cells indicated (right, arrowheads).

DOI: https://doi.org/10.7554/eLife.47328.009

The following figure supplement is available for figure 6:

**Figure supplement 1.** Survival and Plasma Cell Abundance are not altered by ST6GAL1 Expression.
DOI: https://doi.org/10.7554/eLife.47328.010

($r^2$ = 0.42, p=0.0083) (*Figure 6b*). Patients with low ST6GAL1 scores (<100) had higher frequencies of mature neutrophils (22.32 ± 4.11%), whereas those with high ST6GAL1 scores (>100) had markedly lower frequency (9.1 ± 2.6%) (*Figure 6b* inset). Qualitatively, patients with low ST6GAL1 expression in plasma cells had evidence of abundant granulocytes on H and E staining, whereas high ST6GAL1 expressing patients had far fewer visible granulocytes (Plasma cells indicated by arrowheads on DAB stains, PMNs indicated by arrows and arrowheads in H and E of *Figure 6c*).

## Discussion

Effective cross-talk between components of the medullary environment is central to the demand-driven production of different lineages of blood cells. During systemic inflammation, B cells vacate the bone marrow due to the release of TNF-α and downregulation of CXCL12, salvaging space in preparation for the emergency generation of granulocytes (*Ueda et al., 2004*). However, it remains unclear if the departure of marrow B cells merely creates physical space for granulopoiesis or also disinhibits intrinsic granulopoietic processes. Our data support the existence of a paracrine signaling relationship, mediated by B cells via a novel extracellular glycosylation pathway, to influence the generation of granulocytes. This molecular pathway involves the release of catalytically active ST6GAL1 sialyltransferase from B cells, and may contribute to the reciprocal inhibition of granulopoiesis during homeostasis. Our observations add to the body of literature documenting the emerging regulatory relationship between B cells and neutrophils. The newly-defined 'B-helper' subset of neutrophils ($N_{BH}$) can prime marginal zone B cells for antibody production during inflammation by the release of BAFF (*Scapini et al., 2003*; *Puga et al., 2012*). Conversely, evidence suggests that B cells inhibit neutrophil functions in a number of ways, including blocking chemotaxis and initiating β2 integrin-mediated apoptosis (*Kondratieva et al., 2010*; *Kim et al., 2018*). B cell suppression of neutrophilic influx into the liver and spleen is necessary for the resolution of infections with a number of intracellular organisms, where neutrophils can cause damaging local and systemic inflammation (*Bosio and Elkins, 2001*; *Smelt et al., 2000*; *Buendía et al., 2002*). Here, our observations suggest that B cells influence the generation of new neutrophils by regulating glycosylation within the bone marrow, consistent with a separate body of literature documenting the development of excessive neutrophilic inflammation due to ST6GAL1 insufficiency (*Dougher et al., 2017*; *Nasirikenari et al., 2010*; *Nasirikenari et al., 2006*; *Jones et al., 2012*; *Jones et al., 2010*). These include demonstrations of a profound role for extracellular, distally produced ST6GAL1 in muting the transition of granulocyte-monocyte progenitors (GMP) to granulocyte progenitors (GP) (*Dougher et al., 2017*), and that intravenously infused recombinant ST6GAL1 can attenuate demand granulopoiesis in a mouse model of airway inflammation (*Nasirikenari et al., 2019*).

The existence of sialyltransferases within the extracellular milieau, particularly the blood, has been known for quite some time. Upregulation of serum ST6GAL1 during inflammation was attributed to the induction of hepatic expression, leading to the designation of ST6GAL1 as an acute phase reactant (*Kaplan et al., 1983*; *Jamieson et al., 1993*; *Jamieson et al., 1987*; *Jamieson, 1988*; *van Dijk et al., 1986*). However, it was also recognized that in addition to hepatocytes, B cells are sophisticated expressers of *ST6GAL1*, utilizing multiple tissue-specific transcripts during development (*Wuensch et al., 2000*; *Wang et al., 1993*; *Lo and Lau, 1996*). Given the widespread distribution of ST6GAL1-expressing mature B cells within secondary lymphoid tissues, blood, and bone marrow, we hypothesized that this population could be contributing to the extracellular pool of ST6GAL1, thus regulating the sialylation and development of other hematopoietic cells. The observations in this study demonstrate that human B cells can release functional ST6GAL1 and are capable of modifying hematopoietic stem and progenitor cells (HSPC) in co-culture conditions to suppress granulocytic differentiation. Importantly, this effect is attributable directly to the expression of ST6GAL1 in the B cells, as demonstrated by the CRSPR/Cas9 targeted gene knockouts. In vivo, hematopoietic cell-derived ST6GAL1 is a significant modifier of the sialylation of diverse bone marrow cells. Indeed, the limited endogenous expression of ST6GAL1 in hematopoietic stem and progenitor populations may be compensated for by secreted, extracellular enzyme, making the bone marrow microenvironment a distinct niche space for extrinsic sialylation (*Nasirikenari et al., 2014*). In contrast to untreated wild-type mice, we observed striking regional heterogeneity in bone marrow sialylation among *St6gal1*KO chimeras reconstituted with wild-type bone marrow, with higher sialylation generally at the epiphysis and metaphysis of long bones. Interestingly, these sites of high sialylation also

contained high frequencies of IgD+/ST6GAL1+ donor derived mature B cells. In our experiments, hematopoietic cells were capable of independently reconstituting blood levels of ST6GAL1 in *St6gal1*KO mice to baseline WT levels over 8 weeks, and the vast majority of this could be attributed to *ST6GAL1* expression within B cells. These findings clearly implicate cells of the B lineage as major extra-hepatic determinants of blood ST6GAL1, and argue in favor of a tissue-agnostic model of extrinsic sialylation wherein multiple cell types actively secrete enzyme into the extracellular space. Future studies may uncover whether B cells contribute to the elevation of blood ST6GAL1 during inflammation or are regulated by an entirely different set of stimuli.

In the immortalized human B lymphoblastoid lines we examined, aside from the expected 42 kDa soluble form, we also observed a larger 50 kDa form similar in size to the full-length, unclipped ST6GAL1 (*Weinstein et al., 1987*). The 50 kDa form, which appeared to be catalytically active, was the predominant ST6GAL1 form released by the BACE1-negative MM1.S cells. While the nature and mechanism of release of the 50kDa ST6GAL1 form remain to be investigated, its size is consistent with an intact transmembrane domain, raising the possibility of the release of ST6GAL1 associated vesicles. If true, this mechanism may facilitate shuttling of enzyme between specific cell types to mediate extrinsic sialylation (*McLellan, 2009*; *Zech et al., 2012*). Alternately, the 50 kDa form could represent a form of ST6GAL1 processed by signal peptide peptidases, as reported for other glyco-syltransferases, solubilizing the enzyme without significantly reducing its molecular weight (*Voss et al., 2014*; *Kuhn et al., 2015*).

ST6GAL1 has been implicated in a variety of biological processes relevant to the development of disease, particularly in systemic inflammation (*Jamieson et al., 1993*) and metastatic cancers (*Lu and Gu, 2015*). The sialylation of IgG by ST6GAL1 is necessary for the anti-inflammatory effects of IVIG therapy in autoimmune disease, and variations in serum IgG sialylation have been widely associated with inflammatory diseases (*Pagan et al., 2018*; *Nimmerjahn and Ravetch, 2008*; *Biermann et al., 2016*). Human GWAS studies have also associated genetic variation in *ST6GAL1* with IgA nephropathy and flucloxacillin-induced liver damage (*Li et al., 2015*; *Daly et al., 2009*). In epithelial carcinomas, ST6GAL1 expression confers increased resistance to chemotherapy, hypoxia, and nutrient deprivation by promoting a stem-like phenotype, bolstering signaling through pro-survival and pro-proliferative EGFR, HIF-1α, and NF-κB pathways (*Schultz et al., 2016*; *Britain et al., 2017*; *Chakraborty et al., 2018*; *Holdbrooks et al., 2018*; *Britain et al., 2018*; *Jones et al., 2018*). Numerous early studies in both rodent models and humans have also documented concurrent increases in serum protein sialylation and sialyltransferase activity during malignancy, including in multiple myeloma, implying that ST6GAL1 expressing tumor cells are capable of secreting enzyme into the extracellular pool (*Bernacki and Kim, 1977*; *Cohen et al., 1989b*; *Cohen et al., 1993*; *Chelibonova-Lorer et al., 1986*; *Dairaku et al., 1983*; *Gessner et al., 1993*; *Poon et al., 2005*). The functional consequences of fluctuations in blood ST6GAL1 are yet unexplored, especially in malignancy. This current work is the first to examine how cancer-derived ST6GAL1 can perturb the generation of granulocytes, the best documented biologic role associated with extrinsic ST6GAL1 sialylation (*Nasirikenari et al., 2014*; *Nasirikenari et al., 2006*; *Nasirikenari et al., 2019*; *Jones et al., 2010*). Multiple myeloma (MM) was examined because of the natural localization of tumor cells in the marrow, in proximity to nearby healthy HSPCs. ST6GAL1 expression in MM varied dramatically from patient to patient. We observed that the abundance of segmented granulocytes within the marrow was strikingly and negatively associated with the level of ST6GAL1 expression, but not with the overall abundance of multiple myeloma plasma cells. While localized gradients of cytokines, chemokines, and growth factors are already understood to create functional and developmental marrow niche spaces (*Birbrair and Frenette, 2016*), our data now underscore a novel role for extracellular glycan-modifying enzymes such as ST6GAL1 in the marrow hematopoietic environment.

Recent work in our group suggests that extrinsic ST6GAL1 may have a broad ability to coordinate the development and function of multiple immune cell types (*Dougher et al., 2017*; *Irons and Lau, 2018*; *Nasirikenari et al., 2019*). Given the well-documented role of immune cells in cancer, further investigation into the role of extrinsic glycosylation in cancer is merited. The strong negative association between human multiple myeloma ST6GAL1 expression and neutrophil prevalence indicates that tumor-derived ST6GAL1 may dysregulate the development of bystander immune cells in the tumor microenvironment, with a variety of potential implications. At the very least, the ability of cancer-derived ST6GAL1 to disrupt granulopoiesis predicts a diminished capacity for the patient to

combat bacterial infections. Furthermore, the correlation between ST6GAL1 levels and worse patient outcomes in a number of other cancers may be in part due to the extrinsic modification of mature tumor-associated leukocytes. This is consistent with reports that myeloid cell surface α2,6-sialylation diminishes maturation, activation, antigen cross-presentation and anti-tumor immune responses in dendritic cells, for example (*Silva et al., 2016*; *Crespo et al., 2013*; *Cabral et al., 2013*). Collectively, our data hint at biologic effects of ST6GAL1 in cancer that extend beyond the cell-intrinsic modulation of oncogenic signaling pathways, instead being mediated by a novel, extrinsic axis of glycosylation, and galvanized by the growing importance of immune cells in malignancy.

## Materials and methods

### Animal models

The *St6gal1*KO strain has been backcrossed 15 generations onto a C57BL/6J background and maintained at Roswell Park's Laboratory Animal Shared Resource (LASR) facility. The B cell deficient B6.129S2 – *Ighm*[tm1Cgn]/J mouse µMT (The Jackson Laboratory) was used as a donor and recipient in bone marrow transplantation. The reference CD45.1 expressing wild-type strain used was B6.SJL-Ptprc[a] Pepc[b]/BoyJ, in order to distinguish donor cells from recipient mice, which express the CD45.2 allele of the Ptprc locus. For transplantations, mice received 6 Gy whole body gamma-radiation and were rescued with $4.0 \times 10^6$ whole bone marrow cells from a single donor or two donors equally. Mice were euthanized after 8–10 weeks for analysis. Unless otherwise indicated, mice between 7–10 weeks of age were used, and both sexes were equally represented. Roswell Park Institute of Animal Care and Use Committee approved maintenance of animals and all procedures used.

### Antibodies

For immunoblots and immunohistochemistry, anti-ST6GAL1 (R and D Biosystems), anti-β-tubulin (Cell Signaling Technology), anti-PF4 (Peprotech 500-P05), anti-IgD (eBioscience 11–26 c), SNA-FITC (Vector labs) were used. For flow cytometry, SNA-FITC (Vector Labs), biotinylated MAL-II (Vector Labs), anti-Gr1-PE/Cy5 (RB6-8C5), anti-CD11b-BV711 (M1/70), anti-CD45.2-PE/Cy7 (104), anti-CD45.1-PerCP/Cy5.5 (A20), anti-Ly6G-APC (1A8), anti-Ter119-BV510 (TER-119), anti-CD41-BV421 (MWReg30), anti-c-kit-APC/Cy7 (2B8), and anti-Sca-1-PE (D7) (all Biolegend) were used.

### Analysis of cell lines

Human B lymphoblastoid cell lines were grown in RPMI base medium supplemented with 10% heat-inactivated fetal bovine serum. All analyses were performed during logarithmic growth phase, and cell lines were kept in passage for no more than 6 weeks. All cell lines are from Roswell Park Comprehensive Cancer Center repository. Lines MM1.S and HepG2, which are central to the conclusions here, are certified in our laboratory to be mycoplasma-free.

For RNA analysis, cells were washed, pelleted, and resuspended in TRI Reagent (MRC Inc) and RNA extracted according to manufacturer's instructions. 1.0 µg RNA was converted to cDNA (iSCRIPT kit, Bio-rad), and then amplified by qPCR (iTaq Universal SYBR Green, Bio-rad) with intron-spanning primers towards human ST6GAL1 and BACE1. Relative expression ($2^{dCt}$) was calculated in reference to B2-microglobulin. Primer sequences are as follows: B2M: F 5'-GTGCTCGCGCTACTCTC TCT–3', R 5'-TCAATGTCGGATGGATGAAACCC–3'; ST6GAL1: F 5'-CCTTGGGAGCTATGGGACA TTC–3', R 5'-TATCCACCTGGTCACACAGC–3'; BACE1: F 5'-TCTTCTCCCTGCAGCTTTGT–3', R 5'-CAGCGAGTGGTCGATACCT–3'.

For total protein, cells were washed, pelleted, and resuspended in RIPA cell lysis buffer with protease inhibitors, and 5–10 µg of total protein resolved on 10% SDS gels, transferred onto activated PVDF membranes, and blocked in 5% fat-free milk for 1 hr. Blots were probed with primary antibody overnight at 4C, then washed and incubated with HRP-conjugated secondary for 1 hr. Membranes were developed using Pierce ECL WB Substrate (Thermo Scientific) and imaged using ChemiDoc Touch (Bio-rad). For analysis of secreted protein, cells were seeded at a density of $10^6$ cells in 1 ml of serum-free RPMI in 12-well plates. Cell-free conditioned medium was collected after 24, 48, and 72 hr by pipetting and centrifugation at 1,000 rpm to separate cells from supernatant. In order to control for secreted protein per cell, an equal volume (10 µl, 1%) of conditioned media was resolved by 10% SDS-PAGE. In quantification of enzymatic activity, 1.5 µl (0.15%) of total volume was used.

Densitometric quantification of adjusted band intensity was performed separately for 50kD and 42kD forms of ST6GAL1 using ImageJ software.

Sialyltransferase enzymatic activities were quantified by following transfer of $^3$[H]NeuNAc from CMP-$^3$[H]NeuNAc onto the artificial acceptor Gal(β1,4)GlcNAc-o-Bn and separation of the $^3$[H]-trisaccharide products from unreacted $^3$[H]NeuNAc by Sep-Pak C18 reverse phase chromatography. The $^3$[H]NeuNAc- Gal(β1,4)GlcNAc-o-Bn products were further subjected to SNA-agarose chromatography to separate the α2,6-$^3$[H]NeuNAc- (ST6Gal1 product and SNA binding) from the α2,3-$^3$[H]NeuNAc- (SNA flow through) products. This procedure has been described, validated, and utilized previously (*Lee-Sundlov et al., 2017*; *Nasirikenari et al., 2006*; *Nasirikenari et al., 2019*; *Jones et al., 2012*).

## Extrinsic sialylation of fixed hepatocytes

HepG2 cells (ATCC) were seeded at $2 \times 10^5$ cells/ml onto sterile glass cover slips in 6-well dishes for 3 days. Wells were washed with PBS and fixed for 5 min in 5% formalin solution. Cover slips were carefully removed from wells, and subjected to 1 hr treatment with 20 µl/ml bacterial sialidase C (Roche) at 37˚C, followed by incubation with ~20X concentrated Louckes conditioned medium at 10% total volume for 2 hr at 37˚C, in the presence or absence of 100 µM CMP-sialic acid charged sugar donor (EMD Millipore). Cover slips were blocked in 5% BSA for 1 hr, stained with SNA-FITC lectin overnight, washed with DAPI, then mounted onto charged microscope slides in 10% glycerol. During all steps, cells were kept moist by incubation within a water-containing chamber. Fluorescence was visualized immediately using a Nikon Eclipse E600 microscope with EXFO X-cite 120 light source. Spot RT3 camera and Spot Software were used to capture images.

## LK (Lin$^{neg}$ cKit$^{pos}$) cell co-culture

*St6gal1*KO mouse bone marrow mononuclear cells were obtained and enriched for c-Kit+ cells using MACS columns (Miltenyi Biotechnology). Resulting Lin-neg:cKit+ (LK) hematopoietic progenitors (HSPCs) were stained for 20 min at 37C with CellTrace Violet (Thermo Fisher), as per manufacturer's instructions. Stained cells were quenched with media before quantification, and 10,000 cells cultured in 96-well round-bottom plates with either 10,000 or 40,000 human B lymphoblastoid cells at logarithmic growth phase, supplemented with 0.05 mM CMP-Sia (EMD Millipore). To induce differentiation and proliferation, cultures were supplemented with recombinant SCF (50 ng/ml; BioVision), G-CSF (20 ng/ml; Peprotech), IL-3 (5 ng/ml; BioVision), TPO (25 ng/ml; Peprotech), and FLT-3 (30 ng/ml; Peprotech) in a total volume of 200 µl RPMI medium supplemented with 10% FBS. After three days, cells were analyzed by flow cytometry for CellTrace Violet to discriminate between HSPCs and B cells, and murine cells further analyzed for cell surface glycans and expression of granulocyte markers. Flow cytometry data were acquired with BD LSR II flow cytometer and analyzed with FlowJo software.

## Knockout of ST6GAL1 by CRISPR-Cas9

sgRNA was purchased from Genscript (Cr3: CATTCGCCTGATGAACTCTC and Cr4: CAGATGGG TCCCATACAATT). Neon system (Thermo Fisher) was used for electroporation. 1 µg Cas9 (New England Biolabs Inc) was incubated with 1 µg sgRNA for 20 ~ 25 min at room temperature to form the RNP (ribonucleoprotein) complex. 0.1 ~ 0.25 million MM1.S cells was used in each electroporation. Cells were washed twice with PBS and resuspended in buffer R (10 µl), supplied in the Neon transfection kit. Four electroporation conditions were tested (E1: 1600V, 10 ms, three pulses; E2: 1700V, 10 ms, three pulses; E3: 1600V, 20 ms, three pulses and E4: 1600V. 20 ms, two pulses). The editing efficiency was checked 2 ~ 4 days after electroporation by measuring SNA in flow cytometry. Editing was observed under all electroporation conditions, with significant SNA reduction under E2 conditions. SNA-low transfected cells were sorted by fluorescence activated cell sorting (FACS), then analyzed for intrinsic expression of ST6GAL1 at the RNA and protein level to confirm reduction in expression.

## Analysis of bone marrow chimeras

Femurs of indicated bone marrow transplantation chimeras were flushed extensively to obtain cells. Peripheral blood was collected from the retro-orbital venous plexus in citrate-containing

anticoagulant. All samples were subjected to ammonium-chloride-potassium (ACK) lysis buffer in order to remove anucleated cells, then stained with the appropriate combination of antibodies for 20 min, washed, and analyzed by BD LSR II flow cytometer. Data were analyzed with FlowJo software, and donor status of individual cells was distinguished by CD45.1 and CD45.2 staining.

## Histological analysis of whole murine femurs

Femurs were fixed in a paraformaldehyde–lysine–periodate fixative overnight (0.01 M Sodium-M-Periodate, 0.075M L-Lysine, 1% PFA), rehydrated in 30% sucrose in a phosphate buffer solution for 48 hr, embedded in OCT (TissueTek, Sakura), and snap frozen in an isopentane/dry ice mixture (*Nombela-Arrieta et al., 2013*). Whole longitudinal sections (7 μm) sections were obtained using a Leica Cryostat and the Cryojane tape transfer system. Tissue sections were thawed, rehydrated and permeabilized in Tris-buffered saline with 0.1% Tween (T-TBS), blocked with 5% BSA, then incubated with FITC-conjugated SNA lectin (VectorLabs) for 1 hr, then washed prior to incubation with the appropriate primary antibodies. These were followed by the corresponding Alexa Fluor secondary antibodies (1:500, Invitrogen). Fluorescence whole slide imaging was performed on a Nikon Eclipse Ti2. Quantification of SNA density staining and IgD+ cell localization and numeration analysis was executed using Imaris (Bitplane) and Matlab (MathWorks) software. Using consistent parameters for every bone based on their staining intensity, IgD+ B cells were identified and counted. SNA density staining of the marked regions of interest (ROIs) was normalized from different femurs by accounting for total SNA intensity in the whole bone scan. ROIs were selected at random and varied in sizes and SNA staining intensity. ROIs were located in various parts of the bone marrow but consistent with hematopoietic cell localization, and not covering any major vasculature or bony areas. However, SNA intensity within each individual ROI exhibited low variability.

## Histological analysis of human bone marrow

All experiments involving human samples were evaluated and approved by the Institutional Review Board (IRB) prior to their initiation. Banked human biospecimens were provided by the Disease Bank and BioRepository at Roswell Park Comprehensive Cancer Center under protocol BDR 082017. Human multiple myeloma samples collected from treatment-naïve patients were included in the analysis. Samples from 15 treatment-naïve patients were available from the Repository, de-identified prior to transfer, and associated survival and demographic information and pathology reports were provided via an honest broker. Paraffin-embedded sections of bone marrow biopsies were melted at 55C for 1 hr, twice dehydrated in xylene-containing HistoClear (National Diagnostics), then rehydrated in successive ethanol solutions, and heated in Antigen Unmasking Solution (Vector Labs) for 30 min. Slides were blocked in 5% BSA for 1 hr, incubated overnight with anti-ST6GAL1 antibody, then with anti-goat-HRP secondary (R and D Biosystems) for 1 hr. Tissues were then immersed in Impact DAB stain (Vector Labs) for 120 s and rinsed in water for 3 min. Slides were counterstained for 60 s with hematoxylin. Plasma cells were identified under pathologist guidance as cells with radial distribution of heterochromatin and perinuclear clearing corresponding to the lipophilic Golgi apparatus. The presence of malignancy was confirmed by identification of high density sheets or clumps of morphologically similar plasma cells. ST6GAL1 expression was quantified by evaluation of frequency of positively-staining plasma cells and intensity of staining in at least 5 fields of view per specimen typically containing 50 cells each. The frequency of ST6GAL1+ plasma cells was quantified in five random selections of ten cells from five independent images per patient. The average intensity of staining was graded as follows from 0 to 5: 0 – no evidence of staining, 1 – faint staining, 2 – nut brown perinuclear staining occupying <25% of cell, 3 – nut brown perinuclear staining occupying >25% of cell, 4 – dark brown perinuclear staining, 5 – dark brown or black staining occupying the perinuclear and nuclear regions. Bone marrow plasma cells and neutrophils were quantified by pathologist evaluation at time of diagnosis and provided by the Pathology Shared Resource Network (PSRN).

## Statistical analyses

Experiments were conducted with a minimum sample size calculated for appropriate power to detect changes of at least 2-fold ($\alpha = 0.05$, $\beta = 0.80$, SD = 0.5). Raw data are presented in all figures as mean ± SD.

## Acknowledgements

This work was supported by grant R01AI140736 (to JTYL), R01HL089224 (to KMH), K12HL141954 (to KMH and JTYL). The core facilities of Roswell Park Comprehensive Cancer Center used in this work were supported in part by NIH National Cancer Institute Cancer Center Support Grant CA076056. Additional support includes BRI Director's Fellowship Award to MML-S. We would like to thank Jon Wieser for the computational analysis of bone marrow imaging.

## Additional information

### Funding

| Funder | Grant reference number | Author |
|---|---|---|
| National Institutes of Health | R01AI140736 | Joseph TY Lau |
| National Institutes of Health | R01HL089224 | Karin M Hoffmeister |
| National Institutes of Health | K12HL141954 | Joseph TY Lau |
| National Institutes of Health | K12HK141954 | Karin M Hoffmeister |
| National Cancer Institute | CA076056 | Joseph TY Lau |
| BRI Director's Fellowship Award | | Melissa M Lee-Sundlov |

The funders had no role in study design, data collection and interpretation, or the decision to submit the work for publication.

### Author contributions

Eric E Irons, Conceptualization, Formal analysis, Investigation, Methodology, Writing—original draft, Writing—review and editing; Melissa M Lee-Sundlov, Karin M Hoffmeister, Investigation, Methodology; Yuqi Zhu, Resources, Methodology; Sriram Neelamegham, Resources, Supervision, Visualization, Methodology; Joseph TY Lau, Conceptualization, Formal analysis, Supervision, Funding acquisition, Methodology, Writing—original draft, Writing—review and editing

### Author ORCIDs

Melissa M Lee-Sundlov (iD) http://orcid.org/0000-0002-8290-8586
Joseph TY Lau (iD) https://orcid.org/0000-0002-5128-2664

### Ethics

Animal experimentation: Roswell Park Institute of Animal Care and Use Committee approved maintenance of animals and all procedures used, under protocol 1071M.

### Decision letter and Author response

Decision letter https://doi.org/10.7554/eLife.47328.013
Author response https://doi.org/10.7554/eLife.47328.014

## Additional files

### Supplementary files

• Transparent reporting form
DOI: https://doi.org/10.7554/eLife.47328.011

### Data availability

All data generated or analyzed in this study are included in the manuscript and supporting files.

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
