## [Decision Letter]

Thank you for submitting your article "B cells suppress medullary granulopoiesis by an extracellular glycosylation-dependent mechanism" for consideration by *eLife*. Your article has been reviewed by three peer reviewers, including Jamey Marth as the Reviewing Editor and Reviewer #1, and the evaluation has been overseen by Satyajit Rath as the Senior Editor.

The reviewers have discussed the reviews with one another and the Reviewing Editor has drafted this decision to help you prepare a revised submission.

Summary:

Overall the reviewers were in agreement of the general importance of the findings presented, and the potential for a revision of this manuscript to become appropriate for publication in *eLife*. However there were many specific concerns raised by all three reviewers. These comments indicate the need for a significant revision that will also require additional data. It is recommended that the authors address each comment of the reviewers carefully in composing a revised manuscript.

Essential revisions:

Multiple datasets identified in the reviews need additional data points to achieve a more robust statistical significance. In addition, some data appears insufficient and of poor quality, and thus inadequate for review and publication. Various methods used in the manuscript are missing. Data analyzed in the co-culture experiments of Figure 3 should include one or more ST6 KO B cells to determine the B cell effect and considering poor correlations with ST6 RNA and SNA binding (Figure 1/Figure 3). Data presented in the studies of Figure 5 should be further elaborated upon regarding choosing and analyzing ROIs as suggested in the reviews, and the inclusion of µMT donors would be of value. The impact of B cells as presented from data in Figure 6 needs further analysis including cell subsets top gauge the degree of B cell effect versus other cell types present.. The human patient data of Figure 7 needs careful explanation of methods used, as indicated, and better quality histological sections for review and publication.

The individual reviews are provided below so as to provide the authors access to the context and logic for the revisions needed.

*Reviewer #1:*

Irons et al. report ST6Gal1-dependent sialylation of bone marrow progenitor cells results in reductions of Gr1+ granulopoeisis in mice and in corollary cases of human patients with Multiple Myeloma. The authors follow up on their past publications reporting the presence of extracellular sialylation by secreted forms of the ST6Gal-1 sialyltransferase, which is made in a number of tissues and cell types including the liver and B cells, and its role in modulating leukocyte differentiation and inflammatory responses. Cell type specific sources of extracellular ST6Gal1 and its donor substrate among specific studies published have remained mostly unclarified as this enzyme can be found at cell surfaces including among platelets and hepatocytes as well as being secreted into the blood. With the current manuscript, a role of ST6Gal1 in modulating granulopoiesis is the focus. The mechanism involved regarding which proteins are being sialylated and how sialylation modulates granulopoiesis in the bone marrow is not addressed, which is a weakness of the manuscript. At various places the manuscript is difficult to understand and key experimental methods are sometimes lacking. However, there are a number of important findings that with some additional data would provide a strong rationale for publication. The findings of potential involvement of bone marrow megakaryocytes are important, and the authors demonstrate strong linkages of ST6Gal-1 function vis SNA lectin binding with regulation of bone marrow hematopoiesis. The text can be pared down especially Introduction and Discussion sections to achieve a more focused presentation. I have listed other critiques for revision below, including the use of data sets larger than n=2, which would provide me with even greater enthusiasm for publication.

1) The secretion of the 'full length' version of ST6Gal1 may reflect the presence of exosomes in the assays of Figure 1, and is something that might in the future be investigated as exosomes have the potential to transport not just ST6Gal1 but pre-sialylated proteins and donor substrates, in trans. The authors do not comment or reference this possibility. In that regard, the reference to the use of "intracellular proteins" in the Figure 1 legend and in the Materials and methods is incorrect. Total cellular proteins appear to have been analyzed (B and C), including cell surface proteins, and which may have included exosomes from the supernatant, depending on the method of protein isolation, including the centrifugal forces used. The method of protein isolation is absent from the Materials and methods.

2) The studies of HSPCs as presented in Figure 3B could be strengthened by providing more than 2 data points per condition. Key data presented in Figure 3C cannot be read as the text is blurry and too small. So it is not clear if this particular data point is represented in Figure 3D. I also suggest the inclusion of 0:1 co-culture condition data for Figure 3C, along with a clearer depiction and presentation of the gating of cell populations and their frequencies in these examples, and some description of the double-positive population (Gr1+/CD11b+) in the text, which is likely also affected by changes to abundance of Gr1+/CD11b- cells.

3) Partial reconstitution of the bone marrow compartment in chimeras was used to assess the trans sialylation of multiple cell types in vivo. This approach is effective as presented but has only 2 data points (Figure 4B and 4D). The use of only two data points at various places in the manuscript, while technically allowable as a bare minimum, becomes much less effective and compelling when including data involving animal models with individual variation.

4) The authors present an in-situ bone marrow analysis of chimeras to acquire data linking higher levels of ST6Gal1 function (detected by SNA binding), to regions of the bone marrow that are also high in abundance of donor-derived B cells and megakaryocytes. These data are presented in Figure 5. It is not clear if the same cell numbers are being analyzed in each chosen ROI. That data should be provided. It would also be valuable to have a normal correlate panel of wild-type marrow, which could be used to determine whether megakaryocyte lineage cells are normally approximated with highest SNA binding levels in the bone marrow, which if so would further strengthen the relevance of the findings. The potential for sialylation without bone marrow B cells, as is suggested, can be determined using µMT mutant donors as well as the same double mutant recipient used in the current study. The fraction of non B cell ST6Gal1 sialylation in the bone marrow could then be determined. The Figure 5 term "Full Stain" is undefined and unclear. This study also presents very small data sets from small numbers of ROIs and as indicated in the Figure 5 legend from 2 biological replicates.

5) To determine the contribution of B cells, the authors do a study using bone marrow chimeras, which now includes µMT mutant donors, looking specifically at circulating blood cells 8 weeks following partial reconstitution. The data provide evidence of B cell involvement in providing a small degree of ST6Gal1 sialylation measured by SNA binding among circulating donor-derived blood cells. Controls in this study would be helpful to make this determination more clear and robust, including relevant donors without ST6Gal1. It would also be valuable to have more information regarding sialylation among specific cell subsets in circulation, as perhaps some cell types in this model are more of less affected in their sialylation by B cells. In Figure 6C, it is difficult to determine if the difference presented in the cytometry scans reflect the same number of events/cell numbers, as it looks as if there may be more cells being analyzed in the case of the µMT donor set, while the range of SNA binding is not elevated beyond normal.

6) The data provided by the human patients diagnosed with Multiple Myeloma needs more clarification. It is unclear how the data are being analyzed in Figure 7A, there is nothing in the Materials and methods or legend that addresses this. As with all the histology results in Figure 5, the numbers of sections in area and numbers of cells analyzed in total in Figure 7 can be provided by current histological techniques. The tissue panels in Figure 7D are of insufficient quality to identify neutrophils (arrows) and other basic cellular histological features. Regions of interest including higher magnifications are also needed. In studying human bone marrow, the authors have the opportunity to follow up on the megakaryocyte findings, and the SNA reactivity measurements, which might further tie in their findings in Figure 5 with in-situ human studies.

*Reviewer #2:*

This manuscript provides evidence that ST6GAL1 is secreted from B cells to contribute to serum levels and that B cells in bone marrow, perhaps in collaboration with megakaryocytes, are responsible for adding sialic acid to hemopoietic cell progenitors leading to suppression of granulopoiesis. The data are generally of good quality but numerous specifics are missing. The following points should be addressed:

1) In the Introduction, second paragraph: "terminal glycosyltransfearases" – spelling as well as jargon that would not be comprehended by scientists in general.

2) Subsection” Human B lymphoblastoid cells secrete enzymatically active ST6Gal-1”, first paragraph, "formally studied." Do the authors mean to say formerly?

3) Subsection “Human B lymphoblastoid cells secrete enzymatically active ST6Gal-1”, second paragraph. Single examples of human B cell lines at different stages of development were examined. The data would be more robust if independent cell lines at those same stages gave the same results.

4) "In conditioned medium from day 3 of culture, α2,6-sialyltransferase activity generally agreed with protein analyses, whereas α2,3-sialyltransferase activity varied independently (Figure 1D)." This statement is meaningless unless the authors can explain "protein analyses"

5) Figure 1 Are the secreted 50 and 42 kD forms of ST6Gal1 equivalently active? It may not be a completely full length form that is present in the medium. Do SPPL peptidases play any role in cleaving either form? How was the Western blot in Figure 1C imaged – it has a non-specific black blob covering the ST6Gal-1 band in the last lane.

The histogram in Figure 1C appears to represent only the blot in Figure 1C. How reproducible are these data? A histogram including results of several blots should be shown.

What amount of conditioned medium was assayed to obtain the data in Figure 1D ? How was the experiment performed in terms of samples collected at different times, replicates etc.? The description and methods are too brief to enable replication.

It is not acceptable to infer that the remaining activity was due to ST3GAL activity and to label the figure as though that fact was established.

6) Figure 2. Details of methods, amounts, timing and replication are needed in the legend and Materials and methods.

The cellular localization of the SNA signal does not seem consistent with sialidase acting on the surface of fixed cells, nor with SNA apparently inside fixed cells.

7) Details of experimental methods to set up co-cultures and to analyze by flow cytometry, numbers of cells co-cultured, culture conditions, concentrations of cytokines etc. are needed.

Why were ckit+ bone marrow cells used? Progenitors ae usually selected by a more elaborate protocol involving lineage depletion and Sca1.

8) Figure 4 Show a2,3sialyltransferase (SiaT) activity analysis and describe how this specific activity was assayed. Do the levels of the a2,3 enzyme also rise to those found in serum from WT mice? How many chimeras were assayed?

What was the efficiency of chimerism? Were the numbers of hemopoietic cells of each subtype similarly represented? These data should be included in the manuscript, preferably in Figure 4.

What does n=5 relate to in Figure 4D? Is each symbol representative of a mouse?

9) Figure 5. The methods for quantitating femur co-localization data need to be more completely described. Were the ROI data obtained from each femur of each mouse? The correlations would be more robust with more data, especially of high SNA ROIs.

10) Figure 6B cells are clearly not the only source of serum a2,6SiaT. This should be discussed.

How many mice were used to generate Figure 6B versus 6C?

11) Figure 7. The bone marrow H&E from the SNA-low patient is poor. Another example should be shown.

12) General.

Complete information on each Ab used should be given, including clone numbers.

How was non-mitogenic FCS prepared?

Define ACK lysis.

Nomenclature should conform with HGNC convention of ST6GAL1 for protein, St6gal1 (italicized) for the mouse gene, and ST6GAL1 (italicized) for the human gene.

*Reviewer #3:*

In this manuscript, the authors describe the ability of B cell-lineage cells to release the sialyltransferase ST6Gal1, which may contribute to the 'extracellular glycosylation' reported by the corresponding author and others previously. While there are a number of tantalizing observations herein, the issues with the data shown and the conclusions provided reduce enthusiasm to this contribution to a very important field of study. For example, it is not clear that the data in Figure 7 has anything to do with the rest of the manuscript. Also, in several cases, the number of biological replicates was 2, with only 1 being shown, which is inadequate. Further detailed comments are below:

- In Figure 1, the authors are using cancer/immortal cell lines to assess ST6Gal1 and BACE1 expression, but the relationship of these values to normal primary cells in vitro and in vivo is unclear. As pointed out by the authors, cancer is associated with robust changes in glycoforms. Moreover, culturing in standard media contains more glucose than what is normal in vivo, which is also associated with changes in glycosylation. These data have little value in understanding the biology of ST6Gal1 in B cells in the absence of studies using primary cells (human or mouse) for confirmation.

- In Figure 1, the authors also make generalizations about the relative expression levels in B cells from different anatomical locations/developmental stages. But again, these are cancer/immortalized cell lines. Such connections require much more data and multiple cell lines, not just one, for each stage being represented.

- Also in Figure 1, the authors state that the Western blot "confirms" the mRNA levels, but this is a mischaracterization. The protein level (Figure 1B) does not mirror the mRNA level (Figure 1A). Moreover, the protein level was seemingly measured only once, hence no error bars in the graph (Figure 1B). The Western blots are also missing visible size standards.

- Figure 1C is missing repeat experiments and statistics.

- Figure 1D is missing controls with 2,3 and 2,6 recombinant sialyltransferases to demonstrate the rigor of the assay, and the completeness of the SNA capture. The authors also describe the SNA-agarose precipitation as "column chromatography", which is somewhat of a mischaracterization.

- In Figure 2, the images are nice, but a measure of fluorescence intensity of the cells under the varied conditions (and with biological replicates) using flow cytometry would provide statistical power and quantitation.

- Figure 3 is confusing. First, the amount of change in SNA staining does not correlate with the amount of ST6Gal1 being produced by each B cell line (compare changes in Figure 3B with the protein levels in Figure 1B). This calls into question whether the effect of the B cells is truly because they produce ST6Gal1, or if their presence in the culture alters the metabolic state of the HSPCs such that their glycosylation changes. Indeed, one can see a decrease in a2,3-Sia with the RPMI8226 cells, even though they are releasing an a2,3-Sia transferase (Figure 1D). Similarly, the MM1 cells seem to produce a lot of a2,3-Sia transferase activity, and yet no change in glycosylation is seen in the HSPCs. Does this mean that these STs cannot add Sia to the cells? The authors should demonstrate a dependence on ST6Gal1 in these assays by knocking out ST6Gal1 in each line and repeating the experiment.

- In Figure 3D, is it possible that the anti-GR1 antibody binding if impacted by sialylation? Does GR-1 transcription (mRNA) change as well?

- Figure 4B is lacking a comparative control of WT mice not subjected to any bone marrow transfer/chimerism. This is important to provide the relative contribution of the hematopoietic compartment to the total ST6Gal1 activity in the extracellular space (marrow and plasma).

- Figure 5B is lacking statistical power (2 biological replicates). There are too few data points to make a meaningful correlation. Additional repeats should solidify the findings easily. Moreover, and perhaps more importantly, the term "co-localization" is not being applied in a clear way. For example, how close do the cells have to be to be called "co-localized"? Simply within the region drawn on the image? If so, there are cells in Region 1 (panel A) that are closer to Region 2 than they are to the left side of Region 1. Thus, the quantitation here is largely arbitrary and without a specific definition (i.e. within X µm). Finally, it would help if these images were done with Z-axis detail.

- For Figure 6, what are the differences in cellularity between the WT and µMT bone marrow, other than the lack of B cells? Could changes in the cell population that accompany the lack of B cells be playing a role? Also, why would it take 6 weeks for the ST6Gal1 activity to become different, and what happens after 10 weeks, since the curves seem to be converging as they were in earlier weeks?

- In Figure 6, the authors state that the Ighm-/- hematopoietic cells within the blood and bone marrow show increase SNA, but the graph in Figure 6C for the blood indicates a lack of statistical significance in this change. This also highlights another problem. Why is data from only one experiment shown? Multiple biological replicates plotted/averaged together should be shown.

- The connection of the data in Figure 7 to the rest of the manuscript is not clear. There is no functional connection made between ST6Gal1 and the granulocyte observation. In addition, the data in Figure 7D are difficult to see (the H&E). If the number of granulocytes from patients was the readout, flow cytometry or automated hematological cell counting should be performed to provide a more quantitative measure of cellularity.

---

## [Author Response]

Essential revisions:Multiple datasets identified in the reviews need additional data points to achieve a more robust statistical significance. In addition, some data appears insufficient and of poor quality, and thus inadequate for review and publication. Various methods used in the manuscript are missing. Data analyzed in the co-culture experiments of Figure 3 should include one or more ST6 KO B cells to determine the B cell effect and considering poor correlations with ST6 RNA and SNA binding (Figure 1/Figure 3). Data presented in the studies of Figure 5 should be further elaborated upon regarding choosing and analyzing ROIs as suggested in the reviews, and the inclusion of µMT donors would be of value. The impact of B cells as presented from data in Figure 6 needs further analysis including cell subsets top gauge the degree of B cell effect versus other cell types present.. The human patient data of Figure 7 needs careful explanation of methods used, as indicated, and better quality histological sections for review and publication.

Essential changes to the manuscript:

The authors thank the many insightful suggestions from the editor and reviewers. In the revision, we have introduced 2 major new experiments to strengthen the manuscript:

1) To validate further that ST6GAL1 released by the B cells was responsible for modification of HSPC cell surface sialylation and muted Gr-1+ cell generation, 2 CRISR/Cas9 mutants with knocked-out ST6GAL1 were generated and analyzed (new Figure 3F). The CRISPR/Cas mutants lost the ability to suppress Gr-1+ cell generation, relative to the original MM1.S cells.

2) New chimeras were constructed to validate the hypothesis that B cells are the major hematopoietic cells supplying extracellular ST6GAL1 in vivo (Figure 4F and G). ST6GAL1-normal donor hematopoietic cells can restore resting circulating ST6GAL1 activities, but donors without B cells (μMT) are unable to elevate circulating ST6GAL1 activities.

3) Construction and analysis of the new cell lines (part 1) involved new collaborators Yuqi Zhu and Sriram Neelamegham. Hence these individuals are now added to the authorship as authors 3 and 4.

Data sets have also been expanded and descriptions edited. (Please see response to individual reviewers’ comments).

The individual reviews are provided below so as to provide the authors access to the context and logic for the revisions needed.Reviewer #1:[…] 1) The secretion of the 'full length' version of ST6Gal1 may reflect the presence of exosomes in the assays of Figure 1, and is something that might in the future be investigated as exosomes have the potential to transport not just ST6Gal1 but pre-sialylated proteins and donor substrates, in trans. The authors do not comment or reference this possibility. In that regard, the reference to the use of "intracellular proteins" in the Figure 1 legend and in the Materials and methods is incorrect. Total cellular proteins appear to have been analyzed (B and C), including cell surface proteins, and which may have included exosomes from the supernatant, depending on the method of protein isolation, including the centrifugal forces used. The method of protein isolation is absent from the Materials and methods.

We thank the reviewer for bringing this salient point to our attention. We agree that the presence of the full-length form implies that the protein is still membrane-embedded and is consistent with its localization in a vesicle. This might suggest exciting new possibilities in ST6Gal-1 transfer and extrinsic sialylation between different cells. Our observations are consistent with both the 50kDa and the 42kDa forms being catalytically active, but we feel it is beyond the scope of this manuscript for further mechanistic dissection into the origin and nature of the 50kDa form. Results and Discussion have been expanded to address this.

The reference to “intracellular proteins” is replaced with “total cellular proteins”.

2) The studies of HSPCs as presented in Figure 3B could be strengthened by providing more than 2 data points per condition. Key data presented in Figure 3C cannot be read as the text is blurry and too small. So it is not clear if this particular data point is represented in Figure 3D. I also suggest the inclusion of 0:1 co-culture condition data for Figure 3C, along with a clearer depiction and presentation of the gating of cell populations and their frequencies in these examples, and some description of the double-positive population (Gr1+/CD11b+) in the text, which is likely also affected by changes to abundance of Gr1+/CD11b- cells.

We apologize for the low-resolution figure quality in the submitted review draft. All figures have been provided as higher-resolution JPEGs in this submission, and various instances of blurry print were replaced with clearer text for your evaluation. The highest resolution images suitable for publication will be included separately. The data in Figure 3C is from one of the single points shown in Figure 3D. Also, Figure 3C has been updated to include the 0:1 monoculture condition as comparison, wherein it can be appreciated that there is a maximal number of Gr-1+ and CD11b+ cells. In addition, a more detailed explanation of CD11b and Gr1 expression in this experiment and in general is provided in the corresponding Results section for context. We have also included data regarding CD11b-/Gr-1- murine cells, which represent cells that have not differentiated to the extent required for expression of these markers. Here, ST6Gal-1 expressing cell lines are seen to enlarge this population.

*3) Partial reconstitution of the bone marrow compartment in chimeras was used to assess the trans sialylation of multiple cell types* in vivo*. This approach is effective as presented but has only 2 data points (Figure 4B and D). The use of only two data points at various places in the manuscript, while technically allowable as a bare minimum, becomes much less effective and compelling when including data involving animal models with individual variation.*

Similar experiments to Figure 4B and D have been performed by our group in the past and published (Nasirikenari et al., 2010, Jones et al., JBC), and these have clearly demonstrated sialylation by extracellular ST6Gal1 in multiple cell types in vivo. Collectively, our data have demonstrated the importance of non-self ST6Gal-1 expression in cell surface sialylation in a number of contexts. The purpose of Figure 4 was to show that extracellular ST6Gal1 can also come from hematopoietic cells. Figure 4 has been modified to show clearly that the collected data had more than 2 points, specifically in reference to Figure 4B, C and E.

4) The authors present an in-situ bone marrow analysis of chimeras to acquire data linking higher levels of ST6Gal1 function (detected by SNA binding), to regions of the bone marrow that are also high in abundance of donor-derived B cells and megakaryocytes. These data are presented in Figure 5. It is not clear if the same cell numbers are being analyzed in each chosen ROI. That data should be provided. It would also be valuable to have a normal correlate panel of wild-type marrow, which could be used to determine whether megakaryocyte lineage cells are normally approximated with highest SNA binding levels in the bone marrow, which if so would further strengthen the relevance of the findings. The potential for sialylation without bone marrow B cells, as is suggested, can be determined using µMT mutant donors as well as the same double mutant recipient used in the current study. The fraction of non B cell ST6Gal1 sialylation in the bone marrow could then be determined. The Figure 5 term "Full Stain" is undefined and unclear. This study also presents very small data sets from small numbers of ROIs and as indicated in the Figure 5 legend from 2 biological replicates.

The reviewer raised important points. The careful design and description of methods in relatively new analytical techniques are critical to accurate interpretation of results. The Materials and methods section for this experiment has been modified to address the points raised here and by other reviewers, including cell numbers per ROI. As a comparison to our original analysis of WT transplanted marrow into *St6gal1*-KO recipients, which showed patchwork restoration of marrow SNA staining, we did stain a wild-type femur and noted the absolutely homogenous SNA staining. This indicates the sialylation patchiness depends on having a *St6gal1*-deficient recipient animal; WT and µMT mice both have native ST6Gal-1, and all showed diffuse and even SNA staining. These are now included in the figure.

The term ‘full stain’ is removed. The number of biological replicates is 4 animals (Figure 5C, ROIs from each replicate in a different colour). Our data indicate a modest positive relationship between IgD+ B cells and the SNA reactivity of the surrounding cells. To maintain focus of this manuscript, which is on the contribution of B cells to marrow sialylation, we are electing to strike the data associating megakaryocyte from this manuscript. The contribution of megakaryocytes to marrow sialylation will be taken up separately at a later time. Including megakaryocytes will bring up the natural mechanistic question of how megakaryocytes contribute to sialylation, which lies beyond the scope of this manuscript.

5) To determine the contribution of B cells, the authors do a study using bone marrow chimeras, which now includes µMT mutant donors, looking specifically at circulating blood cells 8 weeks following partial reconstitution. The data provide evidence of B cell involvement in providing a small degree of ST6Gal1 sialylation measured by SNA binding among circulating donor-derived blood cells. Controls in this study would be helpful to make this determination more clear and robust, including relevant donors without ST6Gal1. It would also be valuable to have more information regarding sialylation among specific cell subsets in circulation, as perhaps some cell types in this model are more of less affected in their sialylation by B cells. In Figure 6C, it is difficult to determine if the difference presented in the cytometry scans reflect the same number of events/cell numbers, as it looks as if there may be more cells being analyzed in the case of the µMT donor set, while the range of SNA binding is not elevated beyond normal.

In addition to the data presented in the original Figure 6, we have performed a follow-up transplantation using a combination of donors: *St6gal1*-KO, *St6gal1*-KO/μMT, μMT, and WT (C57BL/6) that were used to reconstitute the hematopoietic compartment of *St6gal1*-KO recipients. Under this scheme, the specific contribution of ST6Gal-1 originating from B lineage cells to extrinsic sialylation can be assessed with less ambiguity. The new data involving also donors without ST6Gal-1 and/or without B cells clearly show that reconstitution of hematopoietic compartment with ST6Gal-1-competent B lineage cells is absolutely necessary to raise circulating ST6Gal-1 levels. Lack of B cells but otherwise ST6Gal-1 normal (i.e. μMT donor) is clearly insufficient to restore circulating ST6Gal-1 levels. This new data are now appended to the new Figure 4. The original data, presented as the old Figure 6, used a ST6Gal-1 competent recipient, which allows liver-expressed ST6Gal-1 to contribute to the overall extrinsic ST6Gal-1 pool. The new data, moreover, make the old Figure 6 redundant. Therefore we have removed the old Figure 6 since it adds nothing new.

6) The data provided by the human patients diagnosed with Multiple Myeloma needs more clarification. It is unclear how the data are being analyzed in Figure 7A, there is nothing in the Materials and methods or legend that addresses this. As with all the histology results in Figure 5, the numbers of sections in area and numbers of cells analyzed in total in Figure 7 can be provided by current histological techniques. The tissue panels in Figure 7D are of insufficient quality to identify neutrophils (arrows) and other basic cellular histological features. Regions of interest including higher magnifications are also needed. In studying human bone marrow, the authors have the opportunity to follow up on the megakaryocyte findings, and the SNA reactivity measurements, which might further tie in their findings in Figure 5 with in-situ human studies.

The methods for this experiment have been rewritten to include more detail, particularly in regard to the quantification of ST6Gal-1 expression in myeloma cells and the number of fields of view/cells used to make this determination. Changes have been made to allow the reader to more clearly assess the histological features in H&E stains and appreciate the noted difference in neutrophil abundance. These include an overall increase in the size of pictures, inclusion of an insert wherein individual cellular structures can be observed, and clear indications of both neutrophils and plasma cells for the reader to qualitatively assess the correlation. Although we appreciate the suggestion of furthering the investigations into megakaryocytes in the context of the human samples, we believe that this approach would be unlikely to yield meaningful results for several reasons. Firstly, megakaryocytes are well-documented to participate in niche maintenance for bone marrow plasma cells and have a pro-tumorigenic role in multiple myeloma (Yaccoby et al. Blood 2005, Takagi et al. Blood 2015). Thus, any correlative analysis of megakaryocyte abundance or localization should have to consider the confounding effects of megakaryocytes on the tumor itself. Secondly, the DAB staining process only allows analysis of one histological parameter at a time, making the co-identification of megakaryocytes and sialylation difficult.

Reviewer #2:[…]) In the Introduction, second paragraph: "terminal glycosyltransfearases" – spelling as well as jargon that would not be comprehended by scientists in general.

This has been corrected. Thank you.

2) Subsection” Human B lymphoblastoid cells secrete enzymatically active ST6Gal-1”, first paragraph, "formally studied." Do the authors mean to say formerly?

This has been corrected. Thank you.

3) Subsection “Human B lymphoblastoid cells secrete enzymatically active ST6Gal-1”, second paragraph. Single examples of human B cell lines at different stages of development were examined. The data would be more robust if independent cell lines at those same stages gave the same results.

The purpose of our survey of human B cell lines was not to establish a pattern of ST6Gal-1 expression between different stages of development. Such an analysis in primary cells was recently done by us in a separate publication (Irons et al., 2018), and in the B lymphoblastoids much earlier (Wuensch et al., 2000). Our purpose here was to select B-lineage cells that releases differing levels of ST6Gal-1 in order to assess how B-cell derived ST6Gal-1 can influence extrinsic sialylation of nearby cells. While the panel of B cell lines happened to represent different developmental stages, we cannot and do not infer correlations to stages of development based on this very limited sampling.

4) "In conditioned medium from day 3 of culture, α2,6-sialyltransferase activity generally agreed with protein analyses, whereas α2,3-sialyltransferase activity varied independently (Figure 1D)." This statement is meaningless unless the authors can explain "protein analyses"

We have modified this statement to specifically indicate that the reference is to the quantitation of secreted ST6Gal-1 protein from immunoblot analysis of conditioned medium.

5) Figure 1 Are the secreted 50 and 42 kD forms of ST6Gal1 equivalently active? It may not be a completely full length form that is present in the medium.Do SPPL peptidases play any role in cleaving either form?How was the Western blot in Figure 1C imaged – it has a non-specific black blob covering the ST6Gal-1 band in the last lane.The histogram in Figure 1C appears to represent only the blot in Figure 1C. How reproducible are these data? A histogram including results of several blots should be shown.What amount of conditioned medium was assayed to obtain the data in Figure 1D ? How was the experiment performed in terms of samples collected at different times, replicates etc.? The description and methods are too brief to enable replication.It is not acceptable to infer that the remaining activity was due to ST3GAL activity and to label the figure as though that fact was established.

The sialylated Gal(β4)GlcNAc-o-Bn products have been extensively characterized in the past. We have only seen Sia(α2,6)- and Sia(α2,3)- trimers synthesized from blood-borne enzymes. The characterization includes HPLC and MS verification. While the former is retained by SNA-agarose, the latter is not. Thus based on our prior published characterization, we feel comfortable in stating that the SNA-agarose non-binding fraction represents only Sia(α2,3)-trimers. The Materials and methods section now includes bibliographic references to the analysis, particularly Lee-Sundlov et al., 2017.

To the best of our knowledge, both 50kD and 42kD forms are enzymatically active, as evidenced by the detectable sialyltransferase activity in the conditioned medium of MM1.S cells, which express minimal BACE-1 and largely secrete the 50kD form. Although the ability of these various cell lines to extrinsically sialylation HSPC in co-culture conditions varied, we cannot currently attribute such differences to associated changes in protein size.

We appreciate the insightful reference to aminopeptidases, as previous reports have indicated their involvement in the proteolytic processing of ST6Gal-1. Along with the comment regarding exosomes of reviewer 1, we have included an expanded discussion of the possibility for an aminopeptidase-processed ~50kD soluble form in the Discussion.

All immunoblots were imaged using a Biotek ChemiDoc Touch machine for between 30-120 seconds. We believe the “blob” in the last lane of Figure 1C may be partially due to a distortion in that well during gel loading. However, we have also noted in repeat analyses that the 50kD form of the secreted MM1.S ST6Gal-1 often appears as several bands in close proximity, which we have interpreted as multiple forms of similar size with differences in proteolytic or post-translational modification. Please see the included image of the MM1.S cell line secretions with other multiple myeloma cell lines for comparison, where this can be appreciated (see Figure 2—figure supplement 1B and C). We have repeated the experiment to generate histograms with the averages and variation in these samples.

In Figure 1D, parallel cultures of 10^6 cells were seeded in 1ml of serum-free medium in 12 well plates. At 72 hours post-seeding, conditioned medium was collected by pipetting and centrifuged to separate cells. A fixed volume of 1.5ul (0.15% of total) was used in the sialyltransferase activity assay. Details and clarification have been included into the Materials and methods section to allow for replication.

We have changed the figure, Materials and methods, and Results section to more accurately indicate that the data show SNA-reactive and SNA-unreactive sialyltransferase activity.

6) Figure 2. Details of methods, amounts, timing and replication are needed in the legend and Materials and methods. The cellular localization of the SNA signal does not seem consistent with sialidase acting on the surface of fixed cells, nor with SNA apparently inside fixed cells.

These details have now been included as suggested. Our interpretation of the images in Figure 2 is that low-intensity, ‘hazy’ SNA reactivity was due to cell surface sialic acid, which was removed by sialidase and restored in the final panel. In contrast, the bright, punctate points that persist throughout all treatments are intracellular, likely corresponding to the Golgi apparatus, wherein nascent glycoproteins are intrinsically sialylated. We have performed similar experiments with other cell types to validate observed trends in SNA reactivity on HepG2 cells. Also, we have demonstrated that Louckes-derived supernatant is able to raise SNA reactivity on beads conjugated to an asialylated protein substrate (asialofetuin). It is unclear how the reviewer is interpreting these images if sialic acid is not present on the cell surface or inside cells, but we would emphasize that HepG2 cells are generally large cells with an extended cytoplasm, consistent with the distribution of SNA staining.

To further address this point, we have performed flow cytometry analysis of HepG2 cells in a similar experiment, gating only on nucleated cells to remove residual debris and any acellular components. The results have been included in the figure and demonstrate that an increase in SNA reactivity on the cells can be detected.

7) Details of experimental methods to set up co-cultures and to analyze by flow cytometry, numbers of cells co-cultured, culture conditions, concentrations of cytokines etc. are needed. Why were ckit+ bone marrow cells used? Progenitors ae usually selected by a more elaborate protocol involving lineage depletion and Sca1.

We have added detail and clarification to the methods for this section. The culture of c-kit+ hematopoietic cells is routinely performed in the presence of recombinant ST6GAL1 to inhibit the production of granulocytes. We have previously tested and compared various conditions of cell isolation in this assay, including using lineage depleted bone marrow, c-kit+ bone marrow, and lineage-depleted, c-kit+ bone marrow. We have found that the purity of the LK isolation is comparable among these methods, and typically use only c-kit conjugated magnetic beads to isolate the relevant cells. Further, we have successfully reproduced the suppression of granulopoiesis with recombinant ST6GAL1 in cultures of either Lin-/c-kit+ and c-kit+ cells. We also would note that Sca-1+ cells would exclude the granulocyte/monocyte progenitor (GMP), which we have previously identified as the relevant cellular target of extrinsic sialylation to inhibit granulopoiesis.

8) Figure 4 Show a2,3SiaT activity analysis and describe how this specific activity was assayed. Do the levels of the a2,3 enzyme also rise to those found in serum from WT mice? How many chimeras were assayed?What was the efficiency of chimerism? Were the numbers of hemopoietic cells of each subtype similarly represented? These data should be included in the manuscript, preferably in Figure 4.What does n=5 relate to in Figure 4D? Is each symbol representative of a mouse?

The a2,3 SiaT activity has been added to the figure and clearly indicate that levels are steady in the blood after bone marrow transplantation. In addition, we have included resting wild-type average + SD values for both a2,6 and a2,3 sialyltransferase activity for reference (these are 0.1764 + 0.044 and 1.368 + 0.270, respectively). These values are derived from a total of ten mice from three independent experiments. Please note that a2,3 activity that was measured was restricted to the formation of Sia(α2,3)-Gal(β1,4)GlcNAc- from the SNA-unbound fraction of total sialyltransferase activity. The protocol using Gal(beta4)GlcNAc-o-Bn has been used for the past two or more decades, and the nature of the sialyl linkage, e.g. α2,3 or α2,6, with respect to SNA reactivity has been structurally validated previously (e.g. Lee et al., 2014). Blood-borne enzymes construct no other sialyl forms (aside from α2,3 or α2,6) has been found on Gal(β4)GlcNAc-o-Bn, based on HPLC and MS structural product analysis (Sundlov-Lee et al., 2017).

Data shown in Figure 4B were originally the mean representing 4-5 individual chimeras in each group, pooled into smaller subsets of 2-3 animals. Data from individual animals have now been generated and shown in the figure for clarity. The chimerism of host-derived CD45.2+ cells was comparable within the BM of both treatment groups. A complete tabulation of the frequency of the examined host-derived cell types as a fraction of total CD45.2+ cells has been created as well. These data are now included as Figure 4—figure supplement 1, and show that all parameters are comparable between experimental groups. Each symbol in Figure 4 is representative of an individual mouse.

9) Figure 5. The methods for quantitating femur co-localization data need to be more completely described. Were the ROI data obtained from each femur of each mouse? The correlations would be more robust with more data, especially of high SNA ROIs.

The methods have been improved for this section, particularly in regard to the selection of ROIs. In addition, multiple additional biological replicates have been added to this figure in order to increase rigor, including additional high SNA ROIs. As noted in response to reviewer 1, the strength of observed correlations between B cells and SNA reactivity decreased with additional data collection but remains statistically significant. Data for megakaryocytes have been removed from this manuscript as we deemed this lies outside of the focus of the present manuscript, which is about B-lineage cells as a source of extracellular ST6Gal-1 (see our response to reviewer 1)

10) Figure 6B cells are clearly not the only source of serum a2,6SiaT. This should be discussed.How many mice were used to generate Figure 6B versus 6C?

The original data in Figure 6 was generated with n=3 mice per experimental group. The Discussion has been modified to consider the relative contribution of B cells and other cells to the blood level of ST6Gal-1. In response to other questions, we have performed a further marrow transplant experiment using ST6Gal-1 null recipients, and re-establishing the hematopoietic compartment with either B lineage competent/incompetent, and/or ST6Gal-1 competent/incompetent. This new data, presented as the second half of the new Figure 4, clearly established the role of B cells in releasing circulating ST6Gal-1. The old Figure 6 became redundant data, and therefore we have removed it from the present manuscript. Please see our response to reviewer 1 on this matter.

11) Figure 7. The bone marrow H&E from the SNA-low patient is poor. Another example should be shown.

The images have been modified to improve visibility of both the ST6Gal-1 DAB stain and neutrophils. Plasma cells and neutrophils have also been indicated within the images for clarity.

12) General.Complete information on each Ab used should be given, including clone numbers.

Clone number and catalogue number have been added to the Materials and methods section where possible.

How was non-mitogenic FCS prepared?

This was mistakenly written and has been removed. Our FBS from Atlanta Biologicals is not necessarily free of mitogens. We thank the reviewer for catching this.

Define ACK lysis.

This has been rewritten as ammonium-chloride-potassium lysis buffer that selectively destroys red blood cells by osmolysis.

Nomenclature should conform with HGNC convention of ST6GAL1 for protein, St6gal1 (italicized) for the mouse gene, and ST6GAL1 (italicized) for the human gene.

This has been modified throughout the text where specifically appropriate. However, when generic references are made to ST6Gal-1 expression or function, the text has been left as “ST6Gal-1”.

Reviewer #3:[…] While there are a number of tantalizing observations herein, the issues with the data shown and the conclusions provided reduce enthusiasm to this contribution to a very important field of study. For example, it is not clear that the data in Figure 7 has anything to do with the rest of the manuscript. Also, in several cases, the number of biological replicates was 2, with only 1 being shown, which is inadequate. Further detailed comments are below:

*- In Figure 1, the authors are using cancer/immortal cell lines to assess ST6Gal1 and BACE1 expression, but the relationship of these values to normal primary cells* in vitro *and* in vivo *is unclear. As pointed out by the authors, cancer is associated with robust changes in glycoforms. Moreover, culturing in standard media contains more glucose than what is normal* in vivo*, which is also associated with changes in glycosylation. These data have little value in understanding the biology of ST6Gal1 in B cells in the absence of studies using primary cells (human or mouse) for confirmation.*

We acknowledge the reviewer’s critique that the analysis of ST6GAL1 and BACE-1 expression levels does not necessarily extrapolate to primary B cells. However, the goal in this figure is not to characterize the natural expression of these genes in B cell populations (which was performed recently in another publication, Irons et al., 2018), but to determine if cells of the B cell lineage are capable of secreting ST6GAL1, given its expression. The relevance of this observation to in vivo, primary B cells is tested in Figure 4, wherein the presence or absence of B cells in donor bone marrow is seen to directly alter blood ST6GAL1 and non-self cell sialylation. Furthermore, although we agree that glucose in standard media can alter the production of glycan sugars, it is less clear to us how this would alter the expression of sialyltransferases.

- In Figure 1, the authors also make generalizations about the relative expression levels in B cells from different anatomical locations/developmental stages. But again, these are cancer/immortalized cell lines. Such connections require much more data and multiple cell lines, not just one, for each stage being represented.

We have scaled back any claims regarding the relative expression levels in B cells at different developmental stages. Our goal is to evaluate ST6Gal-1 secretion in cells of the B lineage in general, and we selected multiple cell lines at multiple stages of origin in order to present an unbiased sampling. However, we appreciate the reviewer’s valid critique. Given the later focus on multiple myeloma and bone marrow plasma cells, we have analyzed two more plasma cell stage cell lines (U266 and ARH77), which both express and secrete ST6Gal1 enzyme. Thus, in our sampling, 3 of 4 plasma cell lines express and secrete enzymatically active ST6Gal1.

- Also in Figure 1, the authors state that the Western blot "confirms" the mRNA levels, but this is a mischaracterization. The protein level (Figure 1B) does not mirror the mRNA level (Figure 1A). Moreover, the protein level was seemingly measured only once, hence no error bars in the graph (Figure 1B). The Western blots are also missing visible size standards.

We have more accurately described the results and refrained from the language pointed out by the reviewer. We have also repeated the data in multiple immunoblots in order to include variation in the data, as suggested. The size standards have been enlarged and bolded for visibility.

- Figure 1C is missing repeat experiments and statistics.

We thank the reviewer for bringing this salient point to our attention. We have observed this on a number of occasions, and we showed a representative sample. Moreover, we feel that the activity of the released ST6GAL1 (Figure 1D) is more relevant to the central topic area of this report. While we agree that the presence of the full-length form implies that the protein is still membrane-embedded and is consistent with its localization in a vesicle. This might suggest exciting new possibilities in ST6Gal-1 transfer and extrinsic sialylation between different cells. Our observations are consistent with both the 50kDa and the 42kDa forms being catalytically active, but we feel it is beyond the scope of this manuscript for further mechanistic dissection into the origin and nature of the 50kDa form. Results and Discussion have been expanded to address this

- Figure 1D is missing controls with 2,3 and 2,6 recombinant sialyltransferases to demonstrate the rigor of the assay, and the completeness of the SNA capture. The authors also describe the SNA-agarose precipitation as "column chromatography", which is somewhat of a mischaracterization.

The reference to chromatography has been removed. For our sialyltransferase activity assays, we regularly use both blank saline and serum from ST6Gal-1 KO mice as negative controls, on top of which the values given are calculated. Recombinant ST6Gal-1 protein is also used to calibrate the calculation so that a2,3-sialyl product in this reaction is set at zero. In relation to the data presented, the KO serum control is thus set to 0 fmol/min*ul and a stock recombinant rat ST6Gal-1 enzyme gave an activity of 3800 fmol/min*uL at concentration of 2ug/ml. We do not utilize an a2,3-sialyltransferase to verify the specificity of the a2,3-sialyl product or completeness of SNA capture, since the focus of our work is on a2,6-sialylation and the structure captured by SNA lectin has previously been demonstrated by us to be highly specific for a2,6-sialylation (Nasirikenari et al., 2014).

- In Figure 2, the images are nice, but a measure of fluorescence intensity of the cells under the varied conditions (and with biological replicates) using flow cytometry would provide statistical power and quantitation.

We have repeated this experiment with HepG2 cells in suspension and analyzed SNA reactivity by flow cytometry. Average SNA values with biological replicates are now included in the figure.

- Figure 3 is confusing. First, the amount of change in SNA staining does not correlate with the amount of ST6Gal1 being produced by each B cell line (compare changes in Figure 3B with the protein levels in Figure 1B). This calls into question whether the effect of the B cells is truly because they produce ST6Gal1, or if their presence in the culture alters the metabolic state of the HSPCs such that their glycosylation changes. Indeed, one can see a decrease in a2,3-Sia with the RPMI8226 cells, even though they are releasing an a2,3-Sia transferase (Figure 1D). Similarly, the MM1 cells seem to produce a lot of a2,3-Sia transferase activity, and yet no change in glycosylation is seen in the HSPCs. Does this mean that these STs cannot add Sia to the cells? The authors should demonstrate a dependence on ST6Gal1 in these assays by knocking out ST6Gal1 in each line and repeating the experiment.

We acknowledge that there is an imperfect correlation between ST6Gal-1 expression within the B cell lines and the degree of SNA reactivity gained in the murine HSPCs. However, we would bring the attention of the reviewer to several points. Firstly, the increase in a2,6-sialylation only occurred within B cell lines that we have observed to be secreting ST6Gal-1. Secondly, the reactivity towards SNA has been previously reported to be among the most specific lectin-based methods of analyzing glycosylation, with high affinity to a2,6-sialyl-Gal-B1,4-GlcNAc structures, which are only known to be produced by ST6Gal-1 and ST6Gal-2. Thirdly, the ST6Gal-1 deficient HSPCs would be unable to self-sialylate, even if metabolic changes had occurred secondary to the co-culture with B cells. Therefore, we find it unlikely that endogenous sialyltransferase expression within the HSPCs is driving the observed changes in SNA reactivity. However, in order to more rigorously address this, we have generated 2 CRISPR-Cas9 versions of the MM1.S cell line, in which ST6Gal-1 expression was significantly reduced. The data supporting the creation of these modified cell lines is included in the supplementary figures. Data generated from the co-culture of these new cell lines with *St6gal1-KO* LK cells has been added to Figure 3, and demonstrate that dependence on ST6Gal-1 expression for both changes in SNA reactivity and Gr-1 expression.

- In Figure 3D, is it possible that the anti-GR1 antibody binding if impacted by sialylation? Does GR-1 transcription (mRNA) change as well?

We thank the reviewer for this salient question, and have addressed it both here and within the relevant section of the Results. The Gr-1 epitope consists of both Ly6C and Ly6G proteins, both of which are not predicted (by sequence) to contain any N-linked glycosylation sites. Both proteins contain GPI-linked anchors that likely contain glycolipid structures, which ST6Gal-1 is not known to modify. Further, there have been no publications reporting N-linked glycosylation of either protein. Sialylation of IgG can occur on a single N-linked glycan on the Fc portion, which affects Fc receptor binding, but not Fab antigen recognition. Furthermore, altered expression of signaling, gene expression, and acquisition of Gr-1 has been demonstrated in our previous publication detailing the effects of ST6Gal-1 extrinsic sialylation on G-CSF sensitivity in granulocyte progenitors (Dougher et al.).

- Figure 4B is lacking a comparative control of WT mice not subjected to any bone marrow transfer/chimerism. This is important to provide the relative contribution of the hematopoietic compartment to the total ST6Gal1 activity in the extracellular space (marrow and plasma).

We have included the average for 10 adult WT animals from 3 independent experiments in this panel, demonstrating that the hematopoietic compartment can elevate blood a2,6-sialyltransferase activity to levels comparable to (or even higher than) wild-type mice. These average values and standard deviation are shown in blue for reference.

- Figure 5B is lacking statistical power (2 biological replicates). There are too few data points to make a meaningful correlation. Additional repeats should solidify the findings easily. Moreover, and perhaps more importantly, the term "co-localization" is not being applied in a clear way. For example, how close do the cells have to be to be called "co-localized"? Simply within the region drawn on the image? If so, there are cells in Region 1 (panel A) that are closer to Region 2 than they are to the left side of Region 1. Thus, the quantitation here is largely arbitrary and without a specific definition (i.e. within X µm). Finally, it would help if these images were done with Z-axis detail.

We thank the reviewer for these salient comments, and have expanded the data in this figure to include more biological replicates. In addition, the specific methods involved in selecting ROIs, defining colocalization, and quantification. Unfortunately, we have not yet optimized the technology and methods necessary to visualize the Z-axis.

- For Figure 6, what are the differences in cellularity between the WT and µMT bone marrow, other than the lack of B cells? Could changes in the cell population that accompany the lack of B cells be playing a role? Also, why would it take 6 weeks for the ST6Gal1 activity to become different, and what happens after 10 weeks, since the curves seem to be converging as they were in earlier weeks?

The hematopoietic capacity of μMT mice, an established and frequently used line to address a wide range of questions relating to B cell function, is well-documented. Therefore we saw no reason to revisit this question here. Nevertheless, we have also characterized the general hematopoietic populations between WT and μMT marrow, blood, and splenic cells. In general, aside from the reduced B cells, we saw only an increase in T cells. However T cells are an extremely minor population in the bone marrow. Regarding the differences manifesting at 6 weeks, we have observed that reconstitution of the hematopoietic compartment to the stage of the mature B cell occurs approximately at 4 weeks post-reconstitution, so that the ST6GAL1 derived from this lineage may only significantly contribute to blood levels from this point onwards. Existing ST6GAL1 activity in the blood before then likely reflect traditional sources such as the liver. We have performed a repeat experiment with additional controls, as suggested by reviewer 1, and included it as the latter part of the new Figure 4. The old Figure 6 is now redundant data, and given the issues pointed out by the reviewer of ST6Gal1 from other tissues, is now removed.

- In Figure 6, the authors state that the Ighm-/- hematopoietic cells within the blood and bone marrow show increase SNA, but the graph in Figure 6C for the blood indicates a lack of statistical significance in this change. This also highlights another problem. Why is data from only one experiment shown? Multiple biological replicates plotted/averaged together should be shown.

We have performed a similar experiment in St6gal1-KO mice to address reviewer 1, as well as to strengthen the robustness of the original data. The old Figure 6 is removed, and the new data appended to end of as second part of the new Figure 4. Statistical significance issue is addressed.

- The connection of the data in Figure 7 to the rest of the manuscript is not clear. There is no functional connection made between ST6Gal1 and the granulocyte observation. In addition, the data in Figure 7D are difficult to see (the H&E). If the number of granulocytes from patients was the readout, flow cytometry or automated hematological cell counting should be performed to provide a more quantitative measure of cellularity.

We thank the reviewer for pointing out the lack of conceptual connection between Figure 7 and the rest of the work. The observation that extrinsic sialylation influences granulocyte production has been extensively studied in our group over a period of almost fifteen years. The data in this manuscript is not meant to define the mechanism of this phenomenon, but simply to demonstrate a novel context in which it occurs. The quality of the pictures in this figure have been improved. Unfortunately, the samples available in our Institute’s biorepository only contain 15 formalin-fixed pre-treatment samples and do not include banked specimens amenable to cell counting or flow cytometry. Aside from a lengthy and time-intensive prospective collection of patient samples, this kind of analysis would not be feasible. Furthermore, cytology analysis by trained pathologists is the original and gold standard for bone marrow hematologic cell quantification.